# Modeling and simulating the multi-generation product sales, production and inventory system within the context of quality upgrades

Tan Bo[1], Kenan Yuan[2]*, Yirui Ge[3]

1 School of Management Science and Engineering, Dongbei University of Finance and Economics, Dalian, China, 2 School of Finance, Dongbei University of Finance and Economics, Dalian, China, 3 School of Slavonic and East European Studies, University College London, London, United Kingdom

* yuankenan@stumail.dufe.edu.cn

**Data Availability Statement:** All relevant data are within the manuscript and its Supporting Information files.

**Funding:** The research was funded by the National Natural Science Foundation of China, China (Grant No. 72172025); The Zhejiang Province project for

## Abstract

The rapid development of science and technology has led to an increasing number of high-tech enterprises offering new products through successive generations of product upgrades. This trend presents a new challenge for the sustainable operations of enterprises. Based on the Norton-Bass model, this study begins by constructing a multi-generation product diffusion model within a single enterprise in the context of a monopoly under the quality upgrade scenario. Subsequently, a supply model is established based on this foundation, and these two models are seamlessly integrated using product sales volume as an interface, culminating in a comprehensive sales-supply system. This study analyzes the effects of new-product pricing, quality levels, initial stock, and production capacity on the performance of this system. The system dynamics (SD) method was used to simulate and solve the system in the decentralized and centralized decision-making modes, and the two decision-making modes were compared and analyzed. The research reveals several key findings. i) Comprehensive decision optimization yields enhanced profitability through joint optimization calculation of the multi-generation product diffusion system and the supply adjustment system. ii) consumer price sensitivity significantly affects product quality upgrades and profits. A negative correlation exists between consumer price sensitivity and both factors. The upgrades of product quality should be carefully traded off with consideration of pricing and quality costs. iii) Maximizing profits by maintaining a certain order level of backlog or stock shortage is beneficial for overall enterprise profitability. Additionally, optimal production capacity has been identified as a crucial element in efficient operational inventory management. This study expands the multi-generation product diffusion operational theory and provides valuable theoretical support and decision-making foundations for the sustainable management of enterprises.

planning philosophy and social sciences (Grant No. 23NDJC356YB). The funders had no role in study design, data collection and analysis, decision to publish, or preparation of the manuscript.

**Competing interests:** The authors declare no conflicts of interest.

# Introduction

Effective management and strategic planning for the introduction of new products are paramount to the success of nearly all enterprises. The latter realize sustainable growth through the continuous introduction of new products. As single-generation products are unable to meet consumer needs appropriately, many high-tech enterprises adopt multi-generation product upgrades to provide consumers with new products [1]. This can promote repeat purchases by consumers [2] and increase enterprise revenue. Tesla, BYD, Nintendo, Apple, Samsung, and Xiaomi have achieved increased sales and profits by consistently providing consumers with product upgrades. A typical example is Apple, which has advanced development and upgrading technology capabilities and also excellent marketing and supply chain operations. Apple has developed successive generations of products that are launched every year; through the joint operation of sales, production, and inventory, each generation products has always been in a hot state of purchase and obtains high profits [3]. Similar to Apple, Tesla developed a series of model products through multi-generational upgrades. Starting from Model S and progressing to Model 3, there was a gradual improvement in product performance. However, in this multi-generation product upgrade process, new product sales usually create an imbalance between strong consumer demand and Tesla's delayed delivery. Although there will be insufficient production capacity or component supply problems, this situation has repeatedly occurred. Other companies, such as Nintendo, Samsung, Xiaomi, and BYD, exhibit the same phenomenon. This recurring situation proves that these enterprises employ deliberate strategies, suggesting that new products are sold through a closely integrated approach that combines marketing strategies with supply management. Obviously, in the process of multi-generation product operations, whether the enterprise can realize the systematic analysis and planning of sales and supply of new products plays a vital strategic role in the performance improvement and sustainable growth of the enterprise. Although scholars have offered extensive research and analysis on this method of operation, most have only considered the situation of single-generation products or a subset of the multi-generation product scenario. In the case of multi-generation product upgrades, sales and supply decisions for new products will become more complicated. However, there have been insufficient studies in this area.

We identified important factors based on our analysis. We found that the important feature of product upgrading is the quality level and that enterprises can achieve success more effectively by gradually upgrading their product quality across multiple generations rather than attempting leapfrog upgrades [4]. As branding influences product adoption rates through advertising [5], compelling consumers to embrace new products, the enhancement of brand influence is achieved through quality upgrading of multi-generation products. This approach fosters long-term development and reinforces product brand innovation and competitiveness, thus mitigating brand aging. However, these high-tech enterprises face a complex and demanding operating environment in which other factors such as pricing, production capacity, and inventory management also significantly impact their multi-generation product operations. Pricing can significantly influence consumer demand and purchasing behavior; this is a marketing factor that cannot be ignored in enterprises' product operations. To maximize profits, enterprises must also satisfy consumers through production and inventory adjustments. Enterprises face a trade-off between the high cost of maintaining inventory and the shortage cost caused by insufficient supply. Therefore, enterprises must consider the initial stock and production capacity of each product generation stage. In such complicated situations, enterprises must analyze the key factors that affect profits and consider the substitution effect of consumer behaviors between multi-generation products, which jointly play a role in corporate profit and decision-making. The core challenge in this research is to determine how to make

informed decisions regarding pricing, quality, technology upgrades, production, and inventory to optimize profits while maintaining a dynamic equilibrium among these variables. However, existing research has failed to adequately address these challenges. In the context of quality upgrades of multi-generation products, how should enterprises systematically analyze product branding, pricing, quality or technological upgrades, production, and inventory to secure long-term competitive advantages and sustainable growth paths? This is a strategic decision-making problem that enterprises must urgently solve.

Modern research frequently employs computer simulations to study multi-generation product diffusion and supply chains [6–9]. Therefore, we use the system dynamics method to conduct experimental simulations of the system. In this system, we first use the Norton-Bass model to describe consumer demand and the purchasing process by considering two factors: quality level and dynamically declining price. The price factor [10] and the quality level [11] can influence the diffusion process of multi-generation products in marketing. Based on this, we establish a sales and supply (production and inventory) dynamic management system using differential equations. To maximize profit, the system is simulated and calculated under centralized and decentralized decision-making situations; the latter is based on the Stackelberg game process. We analyze the impact of product pricing, quality or technological upgrades, initial stocking, and production capacity on inventory costs and profits; perform an overall optimization calculation; and offer an integrated strategy for sales, production, and inventory.

The main purpose of this research is to develop a multi-generation diffusion model of a single enterprise that integrates a supply system within the framework of sustainable multi-generation product upgrading; explore the pricing, production capacity, and inventory decisions of multi-generation products in the context of quality-level upgrading; and maximize overall profits as the goal of the simulations. This study reveals the intricate relationship between pricing, quality upgrades, inventory, and production capacity, expands the multi-generation product diffusion theory, and provides a theoretical basis for enterprises' new product launches and operational decisions. In other words, this study explores the path to continuous product innovation and improvement for enterprises.

The remainder of this paper is organized as follows. The Literature Review section reviews the literature. The Model section describes the model and the related assumptions. The System section presents the model of an integrated sales and supply system. The Simulation and Experimentation section presents the simulations and experiments of the system based on the system dynamics method. The Sensitivity Analysis section analyzes the sensitivity of the system to variations in each parameter. In the Discussions and practical implications section, we discuss the simulation results and the practical implications of this study. In the Conclusions and future directions section, the paper discusses and summarizes the main contributions of this study and provides possible future research directions.

## Literature review

### Multi-generation products diffusion process

Scholars have conducted extensive research on the subject of multi-generation product upgrades, resulting in a rich literature [5–6, 10–19]. One of the primary methodologies employed in the study of multi-generation product upgrading is the Norton–Bass model [6], which is an extension modeling of new product diffusion based on the Bass model and characterizes consumer demand and purchase processes. The Bass model [20] was further expanded and refined by Norton and Bass to create the Norton–Bass model for the diffusion of multi-generation products [12]. Subsequently, various scholars continued to extend and enhance this model. Jiang et al. [10] developed a generalized multigeneration diffusion model

(Generalized Norton–Bass model) based on the Norton–Bass model. This model separates leap-frogging and switching adoption purchasing behaviors during the diffusion process and substantiates its superiority through an empirical analysis. Some studies [13–16] have incorporated mixed variables into multi-generation models, such as the substitution effect, price, and advertising marketing factors. Other studies have explored various aspects of multi-generation products, such as product planning [17], substitution dynamics between new and old products [18], and consumer strategies [19]. Dhakal [6] conducted a comprehensive review of the multi-generation product diffusion model, summarizing its research methods and application fields.

With the advancement of product diffusion research, studies have incorporated brand influence into the process [11, 21]. Studies have also shown that good brands can send high-quality signals and have a strong relationship with advertising [22, 23]. That is, brand influence serves as an indicator of product quality, which works through advertising, thereby stimulating consumer demand and purchases. Shi [5] built a multi-generation product diffusion model under dual-brand competition based on the Norton-Bass model and conducted an empirical analysis. Pricing plays a pivotal role as an effective tool to mitigate the challenges of high uncertainty associated with the introduction of new products [24]; therefore, it is necessary to introduce price into the diffusion process of multi-generation products. McGtath [25] noted that the experience curve price serves as a defensive pricing strategy to deter potential competitors and operates as a diminishing pricing strategy that can decrease the initial price by 75% over the course of the sales cycle [26]. In addition, quality plays a pivotal role in the entire product diffusion process, as it serves as a metric for gauging product advancement, influencing the cost and pricing of products [27], and subsequently impacting consumers' purchasing decisions. Li [28] studied the interactive relationship between the price and quality of new products using a multinomial logit model and found that the interaction relationship between quality and price encouraged enterprises to provide products with high quality and price through joint optimization of the two. Bala [29] formulated a game-theoretic model to determine the upgrade range and pricing of a product and found that the upgrade cost is a critical factor in determining consumers' purchase decisions. Kim [30] proposes a framework for a cost -benefit analysis that compares the sequential strategy and quality strategy across multiple generations, as well as taking into account inter-generational competition. Tan [11] embedded dynamic price and quality levels into the Norton-Bass model and provided the best marketing strategy in the case of brand competition. Nadar et al. [31] developed a dynamic model based on the Bass diffusion process and suggested that when word-of-mouth is dominant in product diffusion, it is profitable for enterprises to backlog customers. This study introduces quality as an influential factor in the diffusion of multi-generation products. Generally, there is a strong correlation between higher product quality and enhanced word of mouth. Other scholars focused on consumer behavior strategies during product diffusion [16, 32]. Jun [16] studied the selection problem in the process of multi-generation product diffusion and captured the substitution effect between multi-generation products. Kilicay [33] proposes a multi-agent-based model to simulate the cannibalization of multiple generations of product lines. Consumer behavior is reflected in the mutual substitution and cannibalization of the multi-generation product upgrade process.

To summarize, the multigenerational diffusion process of products is influenced by various factors such as brand reputation, product quality, and pricing. These factors interact collectively to determine and impact consumer purchasing decisions. Additionally, it is crucial to consider the substitution effect between different generations of products, as this can significantly influence enterprise decision-making. Given the complexity of managing multi-generational products, a systematic analysis and decision-making approach must be adopted to establish an optimal marketing strategy.

## Production and inventory in the diffusion process

In the process of product diffusion, production and inventory are important factors that affect enterprises' profits. Shen et al. [34] described product demand based on the Bass model and discussed the optimal strategy of price, production, and inventory under capacity constraints; however, they only considered the spread of single-generation products. Kumar and Swaminathan [35] proposed a diffusion model under capacity constraints and provided optimal operational and marketing decisions. Allon and Zeevi [36] determined the pricing, production, and capacity simultaneously, considering the special case of a continuous price decline. Hartl et al. [37] analyzed the planning and decision-making of durable goods companies over continuous time and concluded that the price of new products continues to decline throughout the sales cycle and that companies should increase production capacity as much as possible when introducing new products; however, they did not consider the impact on inventory. Shen et al. [34] explored the impact of order backlogs owing to production capacity constraints on profits by making joint decisions on price, production, and inventory. Their study also discusses stock shortages and expounds on the relationship between stocks and stock shortages. Production capacity also has an important impact on the renewal strategy for multi-generation products [38, 39], and enterprises need to constantly adjust their sales strategies according to their production capacity. Negahban et al. [40] proposed a supply-constrained multigeneration new-product diffusion model and demonstrated an optimal sales and production capacity policy. They also point out the dynamic relationship between initial inventory and production capacity as new products are introduced. Bilginer [41] studied and analyzed the production and sales strategies for two periods of products under limited production capacity, proposed four policies under inventory holding costs, and showed that part of the capacity shortage was beneficial to the enterprise. Li [24, 39] studied inventory planning during the transition period of multi-generation products, focusing on the substitution effect and pricing between old and new products. Nagpal et al. [42] built an EOQ model for successive multi-generation product-diffusion scenarios; however, their model did not include production. Specifically, Ruidas et al. [43] conducted extensive and in-depth research on the multi-generation product renewal problem and proposed a model that incorporates factors such as price, advertising, demand disruption, product substitution, inventory, and production and provides the optimal inventory and pricing strategy under carbon emissions trading. Subsequently, they considered the impact of quality on an updated version of the product, briefly explained the relationship between quality and brand, and developed an EPQ model on this basis [44]. Bayrak et al. [45] considered production recycling in a closed-loop supply chain with multi-generation product diffusion; however, they did not consider the effect of inventory.

Although these studies have extensively researched the diffusion problem of multi-generation products, there are still significant deficiencies in modeling and simulating production and inventory within the Norton-BASS diffusion process, which expands the application scope of the Norton-Bass model in theory. Unlike Ruidas et al. [44], they mainly discuss the quality of the primary product and its impact on the demand for the pro version. We focus on a comprehensive analysis of quality upgrades in multi-generation product diffusion to make optimal sales and supply decisions. We also consider the important roles of quality and price in demand, and our research shows that quality acts on consumers through word of mouth and together with dynamic price. At the same time, insufficient attention has been paid to the characterization of the quality upgrade level and the accumulation of the quality upgrade level in these works, which have a very important effect on the brand reputation and product diffusion. To date, we found no literature that considers these aspects simultaneously.

This study considers that the production capacity and inventory components of the supply system are critical for enterprise profitability and influence both total costs and consumer

**Table 1. Summary of the important literature.**

| Author | Single/multi-generation | Dynamic pricing | Quality level | Brand | Production | Inventory | substitution | Shortage |
|---|---|---|---|---|---|---|---|---|
| Shi [5] | Multi | | | √ | | | | |
| Jiang [10] | Multi | √ | | | | | √ | |
| Tan [11] | Multi | √ | √ | √ | | | | |
| Li [24] | Multi | √ | | | | √ | √ | √ |
| Kim [30] | Multi | | √ | | | | √ | |
| Nadar [31] | Single | | | | √ | | | √ |
| Shen [34] | Single | √ | | | √ | √ | | √ |
| Kumar [35] | Single | | | | √ | √ | | √ |
| Allon [36] | Single | √ | | | √ | √ | | √ |
| Hartl [37] | Multi | | | | √ | | | |
| Schwarz [38] | Multi | | √ | | √ | | | |
| Li [39] | Multi | | | | | √ | √ | |
| Negahban [40] | Multi | | | | √ | √ | √ | √ |
| Bilginer [41] | Multi | | | | √ | √ | | √ |
| Nagpal [42] | Multi | | | | √ | √ | √ | √ |
| Ruidas [43, 44] | Multi | | √ | | √ | √ | √ | √ |
| Bayrak [45] | Multi | | | | √ | | √ | √ |
| Our model | Multi | √ | √ | √ | √ | √ | √ | √ |

demand. Therefore, it is essential to undertake the joint modeling of sales and supply. Existing research mainly focuses on analyzing and optimizing pricing, quality, or technology upgrade strategies within the construction of multi-generation product models and the product upgrading process for sales. Despite the strong relationship between sales, production, and inventory, especially in the context of quality upgrades, there are still deficiencies in the research on the two dimensions of sales and combining supply (inventory and production). Similarly, limited research has examined the significant influence of joint pricing, quality level, production capacity, and inventory on the multi-generation product diffusion process. Therefore, based on the multi-generation diffusion model (Norton-Bass model), this study establishes a dynamic multi-generation product diffusion supply (production–inventory) system model by integrating the factors of brand, pricing, and quality upgrades, which are contained in the sales system. In addition, this study aims to fill these gaps by examining how quality upgrading affects brands, as well as the factors of initial stock and production capacity, which are contained in the supply system. The system model established in this study expands the application scope of the Norton-Bass model in theory and considers the management practice more diverse and comprehensively; a comparison with previous studies in the field of multi-generation product diffusion can be drawn from Table 1.

## The model

### Model assumption

Before constructing the model, we made the following assumptions:

**Assumption (H1).** This paper operates under the premise that it does not account for consumer attrition. In cases where inventory falls short of meeting consumer orders, enterprises request that consumers wait without resulting in customer loss.

**Assumption(H2).** Consumers initiate word-of-mouth discussions about a product after making a payment. As demonstrated by Shen et al. [34] in this paper, even in cases where

orders face delays during the product diffusion process, consumers generate corresponding word-of-mouth feedback immediately after payment.

**Assumption(H3).** Because product quality serves as the foundation for product brand reputation, the quality level of the previous generation products will have an impact on brand reputation [46], which, in turn, affects the innovative adoption of the next generation products [44].

**Assumption(H4).** Owing to constraints related to production costs and other operational expenses, we assume that prices do not decline continuously. We assume that when dynamic pricing reaches a certain point, it remains unchanged until the product is withdrawn from the market [11, 26].

**Assumption(H5).** Referencing the literature [12], this paper assumes that consumers exhibit homogeneity in the multi-generation product diffusion process, disregarding instances of consumer heterogeneity.

## Analysis and expansion of the Norton-BASS model

Next, we analyze the diffusion process of the Norton-Bass model. In the Norton–Bass model [12], the diffusion process of multi-generation products can be expressed as

$$S_1(t) = m_1 F_1(t)[1 - F_2(t - \tau_2)] \tag{1}$$

$$S_2(t) = F_2(t - \tau_2)[m_2 + m_1 F_1(t)] \tag{2}$$

In this model, $m_1$ is the potential volume of the market for the first generation, and $m_2$ is the potential volume of the market for the first and second generations, respectively. $F_i(t)$ represents the diffusion of adoption at time $t$ concerning generation $i$. $f_i(t)$ represents the diffusion rate of adoption at time $t$, $i = 1, 2$. $S_i(t)$ is the sales rate at time $t$ for generation $i$, which represents the market share. This model clearly reflects substitution and competition between the two generations of products in the market.

Consumers behave differently when buying multiple generations of products [32, 47], and it is reasonable to consider the substitution and competition between multiple generations of products as the cannibalization of new products over old products. Therefore, cannibalization between multiple generations of products must be considered in the modeling process [30, 33]. Based on the Norton–Bass model and referring to Jiang [10], this study divides the interaction between multi-generation products into leapfrog adoption and switching adoption, depending on consumer behavior.

The leapfrogging adoptions at time $t$ can be expressed as follows:

$$u_2(t) = m_1 f_1(t) F_2(t - \tau_2), t \geq \tau_2 \tag{3}$$

The switching adoptions at time $t$ can be expressed as follows:

$$w_2(t) = m_1 F_1(t) f_2(t - \tau_2), t \geq \tau_2 \tag{4}$$

In the process of multi-generational product diffusion, second-generation products exert a substitution effect on their first-generation counterparts, which can also be interpreted as market share cannibalization. Jiang [10] categorized cannibalization into two parts: leapfrog adoption and switching adoption. Leapfrog-buying refers to consumers who initially intend to purchase a first-generation product but opt for a second-generation alternative because of its influence. Switching adoption refers to consumers who have already bought a first-generation product but abandoned its use in favor of repeatedly purchasing second-generation offerings

from the same brand. Together, these phenomena create a cannibalization effect across multiple generations of products.

Based on Eq (3) and Eq (4), we add the price and quality factors. Reference [10, 34], we add pricing as a marketing factor to the diffusion process of products, its expression is $e^{-\theta p_1(t)}$. The diffusion of first-generation products is affected by their quality level $k_1$ though word of mouth, while the diffusion of second-generation products is not only affected by their own quality level $k_2$, but also by the quality level of the previous generation, that is, the degree of quality upgrade, that is, $k_1-k_2$. It affects second-generation products through its cumulative effect on the brand [46], and the degree of improvement in quality can be expressed by $\frac{k_2-k_1}{k_1}$. The leapfrogging adoptions at time $t$ are given by

$$u_2(t) = \left(p_1 + [1 + \alpha_k k_1]q_1 \frac{N_1(t)}{M_1}\right)(M_1 - N_1(t))e^{-\theta p_1(t)}\frac{N_2(t-\tau_2)}{M_2} \tag{5}$$

and the switching adoptions at time $t$ are given by

$$w_2(t) = N_1(t)\left(\left[1 + \beta_k\left(\frac{k_2-k_1}{k_1}\right)\right]p_2 + [1 + \alpha_k k_2]q_2 \frac{N_2(t-\tau_2)}{M_2}\right)$$
$$\times \left[1 - \frac{N_2(t-\tau_2)}{M_2}\right]e^{-\theta p_2(t-\tau_2)} \tag{6}$$

$p_i$ is the coefficient of innovation, and the term "advertising coefficient" is also used in certain studies, $i = 1,2$. $q_i$ is the coefficient of imitation, which also represents the "word of mouth coefficient". $k_i$ is the quality level of $i$ generation product. $\alpha_k$ is the influence coefficient of quality on word-of-mouth, and $1+\alpha_k k_i$ represents the impact of quality upgrades on word-of-mouth. Quality can affect word-of-mouth, which can affect sales [40, 48]; the higher the quality of the product, the better the word-of-mouth, which in turn has a positive impact on product proliferation. and we assume $k_2-k_1 \geq 0$; quality is an important factor that can affect brand and reputation. $\beta_k$ is the influence coefficient of quality upgrading on brand, and $1 + \beta_k\left(\frac{k_2-k_1}{k_1}\right)$ represents the impact of a quality upgrade on a brand or advertisement. Because product quality is the basis of product brand influence, the quality of the previous generation products will have an impact on the influence of the brand, which in turn affects innovative users of the next generation products. Product price sensitivities are often intertwined with marketing variable sensitivities, such as pricing. A clear understanding of how price sensitivity changes will help enterprises make better marketing and pricing decisions [49]. $\theta$ stands for price sensitivity coefficient, which reflects the consumer's sensitivity to price, and thus affects the consumer's purchase decision. The potential volume of the market is represented by $M_i$, $i = 1, 2$. $N_i(t)$ represents the cumulative quantity of $i$ generation product diffusion in time $t$. Price as a marketing variable can be incorporated into multi-generation product diffusion models [14, 50], it can be expressed as $e^{-\theta p_i(t)}$. $p_i(t)$ represents the price of the product at time $t$, as the relevant literature has added the price factor to the Norton-Bass model [13, 51, 52], and the pricing strategy we employ is characterized by a dynamic and continuous decline [19, 52]. We assume that $p_i(t) = p_i(0)e^{Rt}$, $p_i(0)$ represents the initial price of a product when it enters the market. $R$ represents the multiplier through which prices decline over time. The first-generation products and the second-generation products are introduced in the market at time $t = 0$ and $t = \tau_2$.

## The system

Referring to the reasoning and calculation process of sales quantity, as in Shen et al. [34], this study builds a systematic dynamic model of the multi-generation diffusion of a single brand

with the goal of maximizing the revenues of the two generations of products. The objective function of the sales subsystem is as follows:

$$Max\pi_s = \int_0^T \left[\frac{dS_1(t)}{dt}(p_1(t) - \tau_{k_1})e^{-rt} + \frac{dS_2(t)}{dt}(p_2(t) - \tau_{k_2})e^{-r(t-\tau_2)}\right]dt \quad (7)$$

To minimize the cost of the supply system, the objective function of the supply subsystem is established as follows:

$$minH_c = \int_0^T \left(c_1 I(t) + c_2 \frac{dB(t)}{dt}\right)e^{-rt}dt \quad (8)$$

$c_1$ represents the cost of inventory per unit quantity, $I(t)$ represents the cumulative number of products at time $t$, and $c_1 I(t)$ represents the carrying cost of inventory at time $t$. $\frac{dB(t)}{dt}$ represents the quantity of stock shortage due to the inability of the inventory system to meet the demand (sales) at time $t$. Because the system designed and constructed in this study prioritizes inventory to meet the accumulated orders or stock shortages, the cost of stock shortage does not increase with time. The cost of being out of stock at time $t$ is $c_2 \frac{dB(t)}{dt}$, where $c_2$ represents the cost of being out of stock per unit quantity.

The two subsystems were merged to create a sale-supply system as follows, and the total profit of the new system is expressed as $\pi_{OM} = \pi_s - H_c$.

$$max\pi_{OM} == \int_0^T \left[\frac{dS_1(t)}{dt}(p_1(t) - \tau_{k_1})e^{-rt} + \frac{dS_2(t)}{dt}(p_2(t) - \tau_{k_2})e^{-r(t-\tau_2)} - c_1 I(t)e^{-rt} - c_2 \frac{dB(t)}{dt}e^{-rt}\right]dt \quad (9)$$

$$\frac{dS_1(t)}{dt} = \left(p_1 + [1 + \alpha_k k_1]q_1 \frac{N_1(t)}{M_1}\right)(M_1 - N_1(t))\left[1 - \frac{N_2(t - \tau_2)}{M_2}\right]e^{-\theta p_1(t)} \quad (10)$$

$$\frac{dS_2(t)}{dt} = [M_2 + N_1(t)]\left(\left[1 + \beta_k\left(\frac{k_2 - k_1}{k_1}\right)\right]p_2 + [1 + \alpha_k k_2]q_2 \frac{N_2(t - \tau_2)}{M_2}\right)$$
$$\times \left[1 - \frac{N_2(t - \tau_2)}{M_2}\right]e^{-\theta p_2(t - \tau_2)} + \left(p_1 + [1 + \alpha_k k_1]q_1 \frac{N_1(t)}{M_1}\right)(M_1 - N_1(t))$$
$$\times \frac{N_2(t - \tau_2)}{M_2}e^{-\theta p_1(t)} \quad (11)$$

$$\frac{dI(t)}{dt} = \delta_1 1_{\{t=0\}} + \delta_2 1_{\{t=\tau_2\}} + \frac{h(t)}{\omega} - \frac{dS_1(t)}{dt} - \frac{dS_2(t)}{dt} - \varepsilon I(t), I(t) \geq 0 \quad (12)$$

$$\frac{dB(t)}{dt} = S_1(t) + S_2(t) - W(t), B(t) \geq 0 \quad (13)$$

$$\frac{dW(t)}{dt} = \frac{dS_1(t)}{dt} + \frac{dS_2(t)}{dt} + B(t), W(t) \geq 0 \quad (14)$$

$$\frac{dh(t)}{dt} = O(P_c \omega\varphi - I(t)) - \frac{h(t)}{\omega}, h(t) \geq 0 \quad (15)$$

where $S_i(t)$ represents the cumulative sales at time $t$. We define the cost of production at the quality level as $\tau_{k_1} = \frac{1}{2}\sigma k_1^2$, $\tau_{k_2} = \frac{1}{2}\sigma k_2^2$. To simplify the complexity of the model, the cost coefficients of the two generations of products were set to the same value. Set the initial stock $\delta_i =$

**Table 2. The interpretation of notation.**

| Notation | Interpretation |
|---|---|
| $M_i$ | Total market size of each generation products |
| $k_i$ | The quality level of each $i$ generation products |
| $p_i$ | Advertising coefficient of each $i$ generation products |
| $q_i$ | Word-of-mouth influence coefficient of each $i$ generation products |
| $p_i(t)$ | Each $i$ generation products dynamic price (price changes) |
| $N_i(t)$ | The cumulative diffusion number of each $i$ generation products at time $t$ |
| $S_i(t)$ | The cumulative sales volume of each $i$ generation products at time $t$ |
| $\pi_s$ | The total sales profit of two generations of products |
| $\tau_2$ | Second generation products launch time |
| $R$ | Diminishing price factor |
| $r$ | Product revenue discount factor |
| $\theta$ | Price sensitivity coefficient of products |
| $\alpha_k$ | The influence coefficient of quality on word-of-mouth |
| $\beta_k$ | The influence coefficient of quality upgrading on brand |
| $B(t)$ | The number of backlogged products |
| $W(t)$ | The total quantity of outbound |
| $\sigma$ | The unit of quality cost coefficient |
| $c_1$ | The unit of inventory cost |
| $c_2$ | The unit of backlog cost |
| $I(t)$ | The number of products of inventory |
| $P_c$ | The capacity of the enterprise |
| $\omega$ | The lead time |
| $\varphi$ | The ordering cycle |
| $h(t)$ | The finished products |
| $\delta_i$ | The quantity of initial stock |
| $\rho_i$ | The initial stock coefficient |
| $H_c$ | The total cost of supply system |
| $\pi_{OM}$ | The total profit of sale and supply system |
| $\varepsilon$ | The loss ratio of inventory |
| $\tau_{k_i}$ | The cost of quality |
| $T$ | Simulation termination time |

$\rho_i M_i$, $\rho_i$ represents the initial stock factor, we set $0 \leq \rho_i \leq 1$. $O(P_c \omega \varphi - I(t))$ = pulseTrain$(0,1,\varphi,$ 150)*$(P_c \omega \varphi - I(t))$,and pulseTrain$(0,1,\varphi, 150)$ represents a multi-pulse function–when a pulse is generated, the function value is 1 when no pulse action function is 0. In this system, we assumed that the production volume was generated from the beginning of the simulation, the pulse duration was one, the pulse interval was one order of magnitude, and the system was always affected during a simulation period of 150 weeks. $h(t)$ represents the number of finished products produced, and $\frac{h(t)}{\omega}$ is the production rate. $\varphi$ is the ordering cycle, and $\omega$ is the lead time. $W(t)$ is the total number of outbound passengers. $\varepsilon$ is the loss ratio of the inventory, this indicates a cause of the imbalance between products production and sales quantity. The notations used are listed in Table 2.

## Simulation and experimentation

The flow chart depicting the SD model, established using AnyLogic software (professional edition 8.8.4), is illustrated in Figs 1 and 2. Fig 1 shows the sales subsystem model under multi-

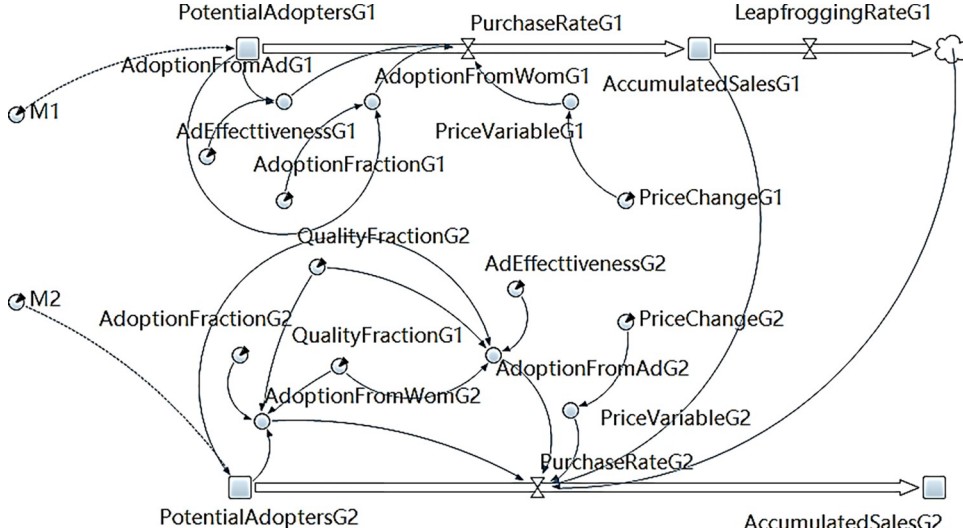

**Fig 1. Sales system model under multi-generation product diffusion.**

generation product diffusion, which describes Eqs (7), (10), and (11). Fig 2 shows the supply subsystem model under multi-generation product diffusion, which intuitively shows Eqs (8) and (12)–(15).

We designed a supply system comprising two parts: production and inventory. Referencing [53, 54], the two subsystems can have two decision modes: noncooperative and cooperative. When the sales and supply subsystems make separate decisions, this situation can be regarded as a non-cooperative mode. We assume that the demand of products determines the supply of products and that the sales system makes decisions first, which in turn affects the demand for products. Next, the inventory system makes decisions based on the state of the sales system. Finally, the sales system determines its decisions based on the current decisions made by the supply system. The decision-making processes of the two systems follow the noncooperative

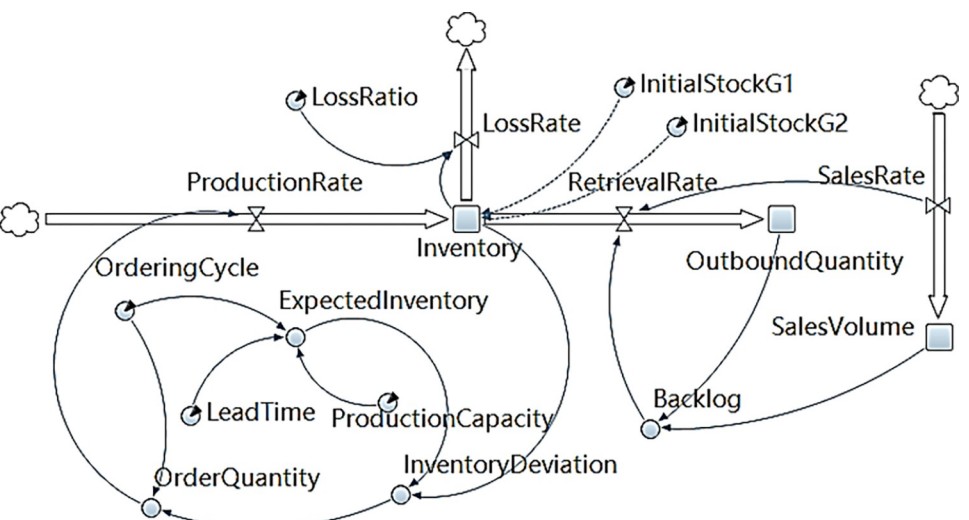

**Fig 2. Supply system model under multi-generation products diffusion.**

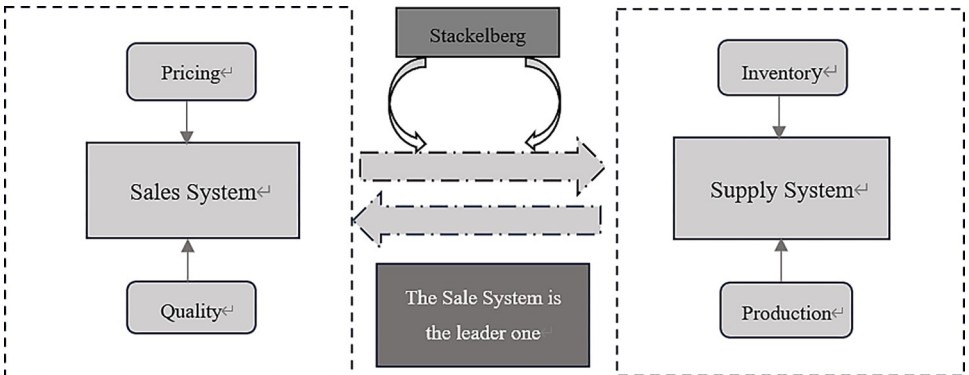

**Fig 3. Stackelberg game decision process of Sales-Supply model system.**

Stackelberg game model [54]. The sales-supply decision process based on the Stackelberg game model is shown in Fig 3.

The basic parameters were set for the basic data in references by Jiang et al. [10], and Krishnan et al. [52], and the other specific basic parameter settings of the model in this study are shown in Table 3.

By changing the parameter assignment, the decision problems of the pricing level, quality level, initial stock, and production capacity were analyzed. Due to the constraints of production costs and other operating expenses, the price did not drop indefinitely. This study assumes that the initial price dynamics decrease to a certain point and remain unchanged until the end of the product sales cycle. This practice is widely used in real-world commercial activities. For instance, Apple often offers a mobile phone product price reduction of 80% from the original price at some point, which accounts for approximately one-fourth of the initial cost, so we set $-0.8 \leq R \leq 0$. $r$ is the discount factor, which is roughly equal to the annual bank rate, and we assume $r = 0.02$; the reason is that this study also considers the difference between the internal and external effects of quality upgrades on the sales system. According to Mitra et al. [55], product quality can only produce 10% of the total utility in the same period but 20% of the total utility in the long-term process. Therefore, this study assumes that the perception coefficient of consumers that word-of-mouth is affected by quality is set to 0.1. The consumer

**Table 3. Basic parameter settings of the system.**

| Parameter | Value | Parameter | Value |
|---|---|---|---|
| $M_1$ | $5.03 \times 10^7$ | $\omega$ | 2 |
| $M_2$ | $21.1 \times 10^7$ | $\varphi$ | 1 |
| $q_1$ | 0.337 | $\rho_1$ | 0.5 |
| $q_2$ | 0.337 | $\rho_2$ | 0.5 |
| $p_1$ | 0.00943 | $\varepsilon$ | 0.0001 |
| $p_2$ | 0.00943 | $c_1$ | 0.1 |
| $k_1$ | 1 | $c_2$ | 0.2 |
| $k_2$ | 1 | $P_c$ | 100,000 |
| $p_1(0)$ | 1 | $\alpha_k$ | 0.2 |
| $p_2(0)$ | 1 | $\beta_k$ | 0.1 |
| $R$ | -0.005 | $\tau_2$ | 50 |
| $\sigma$ | 0.1 | $\theta$ | 0.8 |
| $r$ | 0.02 | $T$ | 150 weeks |

perception coefficient for brand quality improvement was 0.2. Businesses typically release new products once a year, so we assume $\tau_2 = 50$.

## Simulation under decentralized decision-making

First, under the capacity constraint, $P_c = 100{,}000$ is set. This study assumes that a direct demand-oriented sales system plays a leading role and that the enterprise supplies products according to demand. Second, each parameter variable in the fixed inventory subsystem is simulated and calculated for the sales subsystem. To maximize sales revenue, Eqs (7), (10), and (11) were solved and optimized. A genetic algorithm (GA) is used to optimize the overall price and quality of the two generations of products. After 5,000 iterations, the calculation process is shown in Fig 4(A), and the results are as follows:

$$p_1(0) = 1.892, p_2(0) = 2.045$$

$$k_1 = 1, k_2 = 1.391$$

$$\pi_s = 23,653,432.399 \tag{16}$$

Based on the Stackelberg game model, the inventory subsystem optimizes its decision according to the optimal decision of the sales subsystem and simulates the supply subsystem with the goal of minimizing the total inventory cost. Eqs (8) and (12)–(15) are solved and optimized. Similarly, a genetic algorithm (GA) is used to optimize the initial stock coefficient for each product generation. After 5,000 iterations, the calculation process is shown in Fig 4(B).

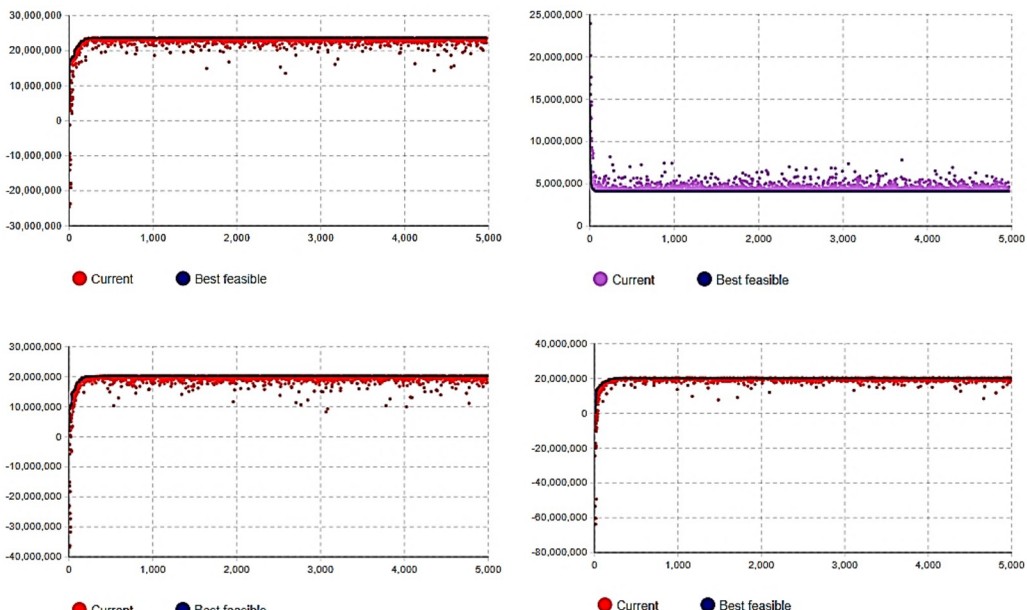

**Fig 4. Optimization process and results of genetic algorithm (GA).** (a) Sales revenue, (b) Inventory costs, (c) Total profit (decentralized), (d) Total profit (centralized).

The optimization results are as follows:

$$\rho_1 = 0.04, \rho_2 = 0$$

$$H_c = 4,136,029.7 \tag{17}$$

Furthermore, the supply system parameters are adjusted based on the optimization. Subsequently, the sales system is optimized based on inventory optimization, aiming to maximize the overall profit of the sales-inventory system, as shown in Fig 4(C). The following results were obtained:

$$p_1(0) = 1.847, p_2(0) = 2.361$$

$$k_1 = 1, k_2 = 1.484$$

$$\rho_1 = 0.04, \rho_2 = 0$$

$$\pi_{OM} = 20,268,825.37 \tag{18}$$

The results above show that if only the first two steps of the distribution are optimized, the total profit can be calculated from $\pi_s - H_c$, and the total profit ($\pi_{OM}$) is 19,517,402.7. After the sales subsystem is optimized in the third step, the profit is significantly improved, resulting in 20,268,825.37. The result after optimization was approximately 3.8% higher, which shows that the distributed optimization method based on the Stackelberg game model designed in this study realizes profit optimization and makes effective improvements.

## Simulation under centralized decision-making

In this section, we carried out a comprehensive optimization of the sales and supply system (Eqs 10–15), that is, we carried out overall planning of sales, inventory, and production, aimed at maximizing operating profit, and optimized the calculation of initial pricing, quality level, and stocking coefficient. The calculation process is shown in Fig 4(D), and the following optimization results were obtained.

$$p_1(0) = 1.825, p_2(0) = 2.347$$

$$k_1 = 1, k_2 = 1.433$$

$$\rho_1 = 0.038, \rho_2 = 0$$

$$\pi_{OM} = 20,274,626.022 \tag{19}$$

The simulation optimization results indicated that the profit of the centralized decision was 0.29% higher than that of the individual decentralized Stackelberg process decision. In contrast to the benchmark model, enterprises upgrade the quality of multiple product generations to maximize their profits. Product pricing will improve because the upgrading of product quality leads to an increase in product development costs, and enterprises must increase their pricing when next-generation products are launched, which is also consistent with the conclusion of the literature [28], which provides high-quality products at high prices. In terms of inventory operations, enterprises use low inventory to reduce inventory costs. Enterprises will adopt a low-inventory approach to reduce inventory costs. When generation products are launched,

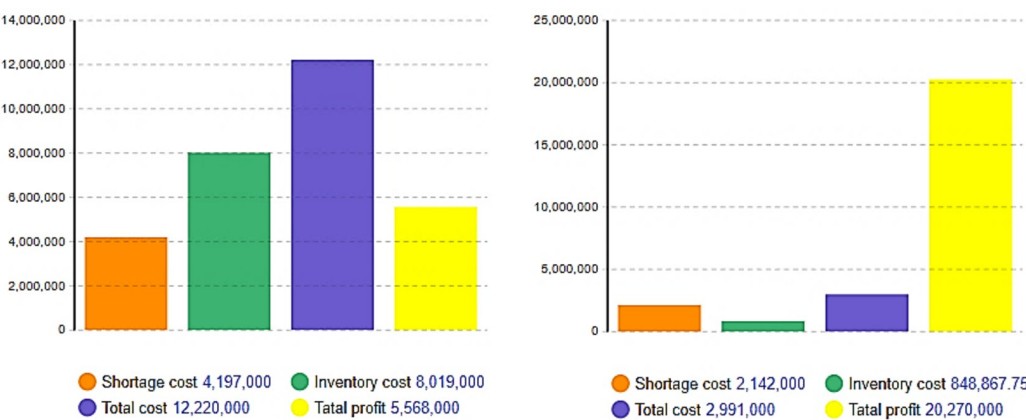

**Fig 5. Simulation results of inventory system before and after optimization.** (a) Before optimization, (b) After optimization. * The blue values represent specific values on the vertical axis.

enterprises even implement a zero-stock strategy, which contributes to overall profitability despite potential order backlogs caused by shortages.

The system was simulated before and after optimization, and the inventory and out-of-stock costs were obtained, as shown in Fig 5.

After optimization, it is evident from the comparison between Fig 5(A) and 5(B) that the total cost decreases and the total profit increases significantly. The total cost decreased from 1,222,000 to 291,000, and the total profit increased from 5,568,000 to 20,270,000. Simultaneously, the shortage level changed from less than the inventory cost before optimization to greater than the inventory cost, as shown in Fig 5(B). This change can be attributed to the reduction in each generation of the products' initial stocking factor, resulting in a decrease in inventory costs from an initial value of 8,019,000 to 848,867.75, consequently leading to a considerable reduction in overall inventory costs. The optimization results revealed that the proportion of out-of-stock costs within the total cost increased from its original value of 34.3% to 71.6%. Nevertheless, by striking a balance between these two factors, there has been an overall reduction in total costs while simultaneously increasing enterprise profitability.

Summary1: The introduction of product upgrades drives companies to increase their pricing, leading them to adopt higher-quality, higher-price strategies.

Summary2: The trade-off between inventory holding costs and out-of-stock costs is a crucial consideration for enterprises in their product operation processes, and maintaining a certain level of out-of-stock or order backlog can contribute to the overall profitability of enterprises.

## Decision optimization under uncertain production capacity

We added production capacity as a variable in the simulation experiment and performed optimization calculations according to the decision-making process under the production capacity constraints above. The results are summarized in Table 4.

**Table 4. Optimization results under the two decision methods.**

| Strategy | $p_1(0)$ | $p_2(0)$ | $k_1$ | $k_2$ | $\rho_1$ | $\rho_2$ | $P_c$ | $\pi_{OM}$ |
|---|---|---|---|---|---|---|---|---|
| Centralized | 1.84 | 2.227 | 1 | 1.471 | 0.089 | 0 | 184,024.718 | 21,223,132.855 |
| Distributed | 1.845 | 2.175 | 1 | 1.444 | 0.104 | 0 | 207,325.736 | 21,179,005.295 |

**Table 5. Optimal decision of enterprises under different quality cost levels.**

| Parameter | $p_1(0)$ | $p_2(0)$ | $k_1$ | $k_2$ | $P_c$ | $\rho_1$ | $\rho_2$ | $\pi_{OM}$ |
|---|---|---|---|---|---|---|---|---|
| $\sigma = 0.02$ | 1.808 | 2.486 | 1 | 4.716 | 203,125.921 | 0.099 | 0 | 24,453,983.282 |
| $\sigma = 0.04$ | 1.823 | 2.339 | 1 | 2.291 | 197,704.076 | 0.097 | 0 | 22,727,531.856 |
| $\sigma = 0.06$ | 1.832 | 2.305 | 1 | 2.181 | 192,112.336 | 0.094 | 0 | 21,968,515.129 |
| $\sigma = 0.08$ | 1.84 | 2.288 | 1 | 1.812 | 182,709.355 | 0.089 | 0 | 21,516,149.674 |
| $\sigma = 0.1$ | 1.84 | 2.227 | 1 | 1.471 | 184,024.718 | 0.089 | 0 | 21,223,132.855 |
| $\sigma = 0.12$ | 1.855 | 2.212 | 1 | 1.228 | 189,065.166 | 0.093 | 0 | 21,006,772.315 |
| $\sigma = 0.14$ | 1.859 | 2.185 | 1 | 1.106 | 189,985.557 | 0.093 | 0 | 20,836,895.463 |
| $\sigma = 0.16$ | 1.864 | 2.173 | 1 | 1.005 | 189,734.771 | 0.093 | 0 | 20,698,482.364 |
| $\sigma = 0.18$ | 1.869 | 2.157 | 1 | 1.000 | 191,122.618 | 0.094 | 0 | 20,563,084.368 |
| $\sigma = 0.2$ | 1.877 | 2.195 | 1 | 1.000 | 187,722.242 | 0.093 | 0 | 20,426,976.162 |

Table 5 demonstrates that enterprises can achieve maximum profits through an overall centralized strategy. Notably, the overall centralized optimization method improves the result by 0.21% compared to the distributed optimization method based on the Stackelberg game model. The results of the initial stock optimization of the two methods show that the additional initial stock of second-generation products is zero, and the enterprise does not need to carry out additional stock. By relying on the current continuous supply mode of the system, the overall profit of the system can be optimized, which also causes second-generation products to present a certain stock shortage in the early stage of sales.

The optimization effects of the two approaches under different capacity constraints are compared below. We normalized all the optimization results, as shown in Fig 6.

The optimization effect of the two methods exhibited an initial increase, followed by a decrease, with an increase in production capacity. Specifically, when the production capacity ranges from 50,000 to 180,000, the centralized optimization yields higher profits, whereas when the production capacity ranges from 180,000 to 250,000, the decentralized optimization leads to higher profits. However, it should be noted that the results of the two approaches are

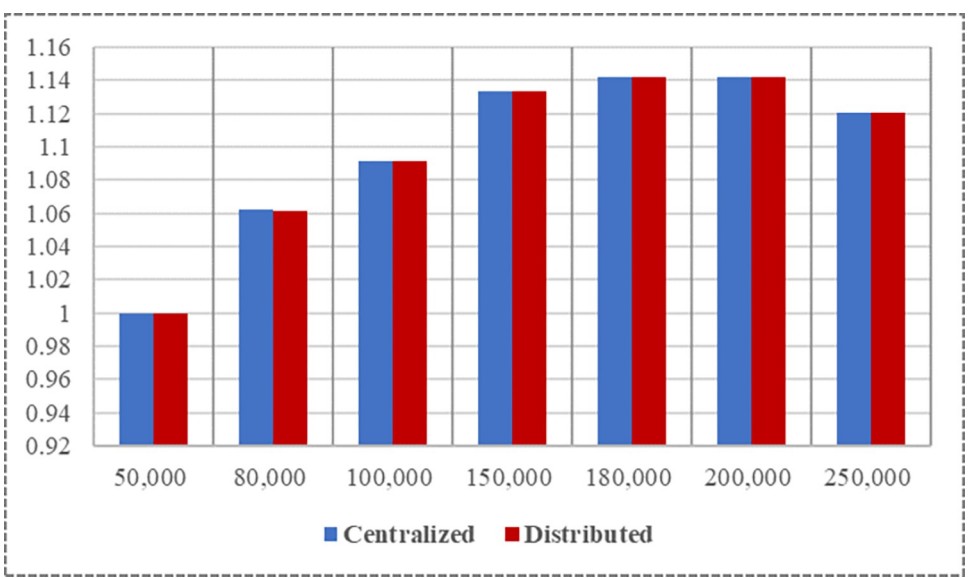

**Fig 6. Comparison of the effects of the two decision-making methods.**

very close. Enterprises need to carefully evaluate various strategies and make informed decisions regarding which optimization strategy to adopt based on their specific circumstances.

Summary3: Through systematic analysis and optimization of the factors affecting the diffusion of multi-generation products, the overall profit of enterprises can be effectively improved.

Summary4: Production capacity constraints significantly influence the optimization strategy adopted by enterprises and serve as a crucial determinant of profitability.

In the process of multi-generation product operation, there will be a certain initial stock when the first-generation product is launched, and then a very low additional or even "zero stock" strategy will be adopted in the second-generation product to achieve the purpose of reducing inventory costs and increasing profits. While inventory costs were reduced, additional out-of-stock costs were incurred. This is the internal reason for adopting the low-inventory strategy. By adjusting the production capacity, the overall profit of the enterprise can be significantly improved. The enterprise must conduct an overall analysis of its sales inventory system to determine its optimal production capacity.

## Sensitivity analysis

Next, sensitivity experiments and analyses were conducted on the model results after centralized decision optimization, including the quality level, initial price, initial inventory, and production capacity. First, we compare the impact on the total inventory cost and overall profit under different second-generation product quality levels, as shown in Fig 7.

In this study, the total inventory cost represents the sum of the cost of holding inventory and the cost of a shortage due to insufficient inventory. The total profit represents the overall revenue from sales minus the total inventory cost. Through the above comparative experiments, it can be observed that the quality of second-generation (G2) products has an important impact on the total cost of inventory in Fig 7, while a total cost of inventory that is too high or too low is not always conducive to the final overall profit. As shown in Fig 7(A), the quality level of the second generation is 1.433, as expressed by Eq (19), and the total inventory cost is not the lowest; however, the overall profit level is the highest, as shown in Fig 7(B). Therefore, enterprises need to conduct a comprehensive analysis of the sales-supply system and make a tradeoff between sales profits and costs when making quality decisions for second-generation products.

The effects of each generation's product pricing on each generation's profit and total profit were then compared and analyzed. The total profit of the product exhibited an initial increase, followed by a decrease, as depicted in Figs 8(A) and 9(A), indicating a correlation with pricing. As shown in Fig 8(B), when the initial price of the first-generation product reaches the optimum, that is, $p_1(0) = 1.825$, the profit of the current-generation products also reaches a maximum, and the change trend also shows an increase and then a decrease. It is noteworthy that

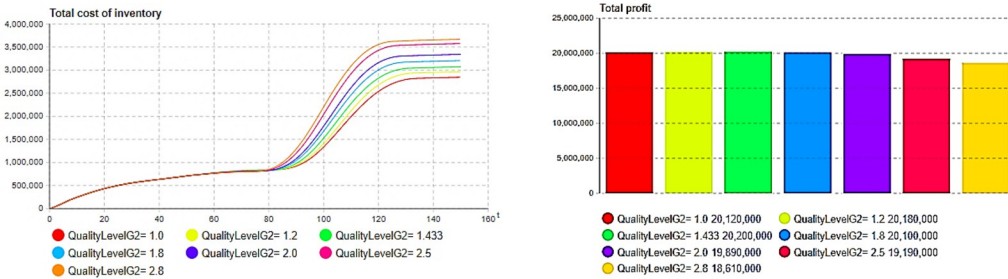

**Fig 7. Impact of quality level upgrade.** (a)Quality: Total cost, (b) Quality: Total profit.

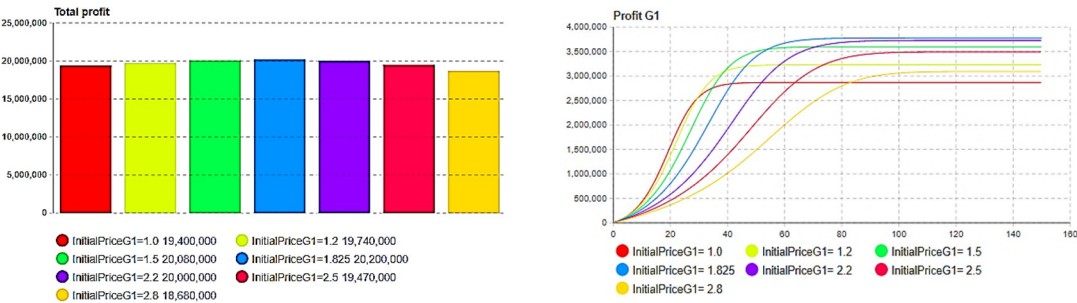

**Fig 8. The impact of the pricing of first-generation(G1) product on itself and the total profit.** (a)Pricing G1 according to total profit, (b) Pricing G1-Profit G1.

the second firm does not attain its maximum individual profit at the optimal initial price ($p_2(0) = 2.347$), as shown in Fig 9(B); however, it contributes to maximizing overall profit, as shown in Fig 9(A). This phenomenon can be attributed to the fact that, under the optimal price, the sales rate of the product reduces inventory-related system response costs, thereby enhancing overall enterprise profitability.

Through simulation in Fig 10(A), when $\rho_1 = 0.038$, the initial inventory point can be found to maximize the total profit. At the same time, there is an optimal inventory point that minimizes the total inventory cost, as shown in Fig 10(B). The total inventory cost is also minimized. As Fig 11(A) and 11(B) show, the initial inventory of second-generation products also shows similar results. The simulation results above show that the initial stocking coefficient of

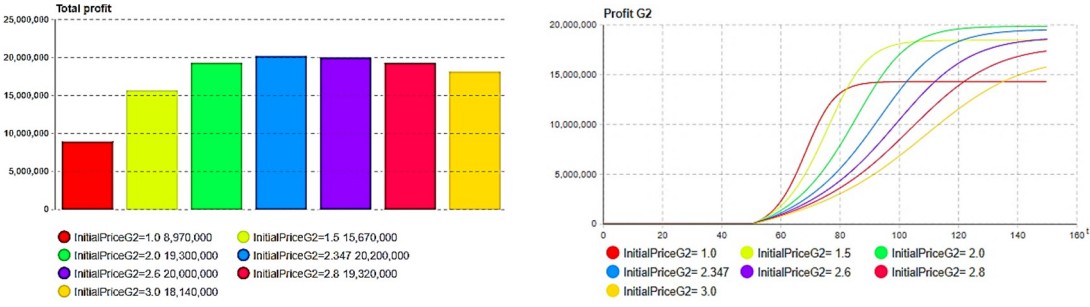

**Fig 9. The impact of the pricing of second-generation(G2) product on itself and the total profit.** (a) Pricing G2 according to total profit, (b) Pricing G1-Profit G1.

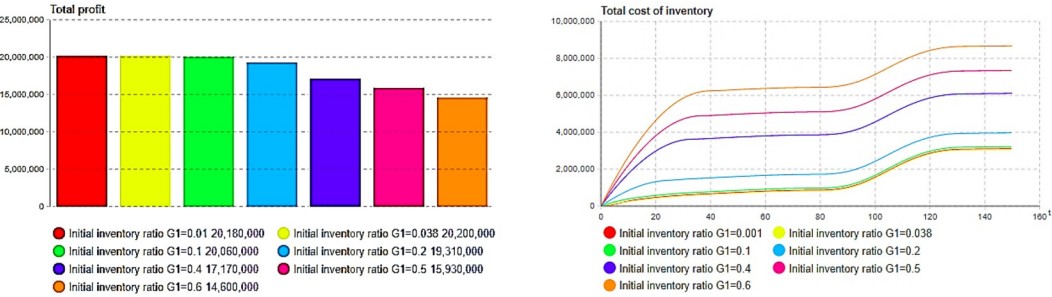

**Fig 10. Influence of stocking factor of first-generation product on profit and inventory cost.** (a) $\rho_1$−total profit, (b) 03 $\rho_1$−total cost.

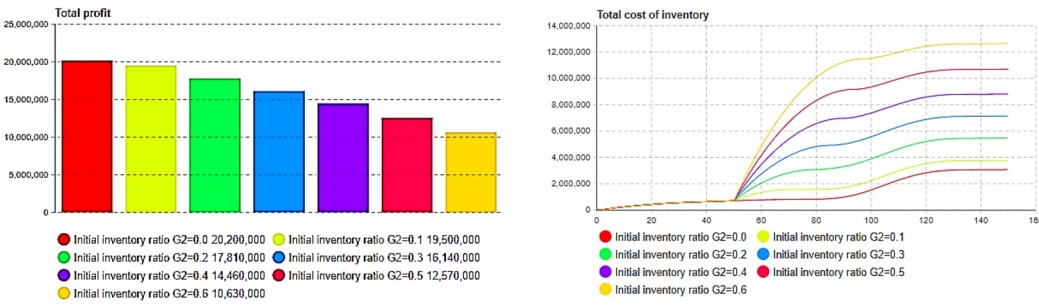

**Fig 11. Influence of stocking factor of second-generation product on profit and inventory cost.** (a) $\rho_2$–total profit, (b) $\rho_2$–total cost.

the product primarily affects the total cost of the product inventory, which, in turn, affects the total profit of the product. It is worth noting that the initial inventory of the second-generation products can be reduced to a very low level, or even "zero stock," which indicates that through dynamic production and replenishment, the overall profit of enterprises can be improved.

By changing the parameters of production capacity to simulate the system, we can see that the maximum profit is observed at $Pc$ = 180,000, as depicted in Fig 12(A), indicating the existence of an optimal production capacity for enterprises, and the results in Table 4 illustrate this point. This underscores the significant impact of production capacity on the total cost. For instance, when $P_c$ = 200,000 in Fig 12(B), the total inventory cost is not minimized. Note that a reduction in out-of-stock costs does not lead to a decrease in total inventory costs. As illustrated in Fig 12(C), with an increase in production capacity, the out-of-stock cost of inventory continues to decrease, whereas the holding cost increases in Fig 12(D). Clearly, when the production capacity continues to increase, consumer demand can be met in a timely manner. Although the cost of a shortage can be reduced, this leads to an increase in inventory costs.

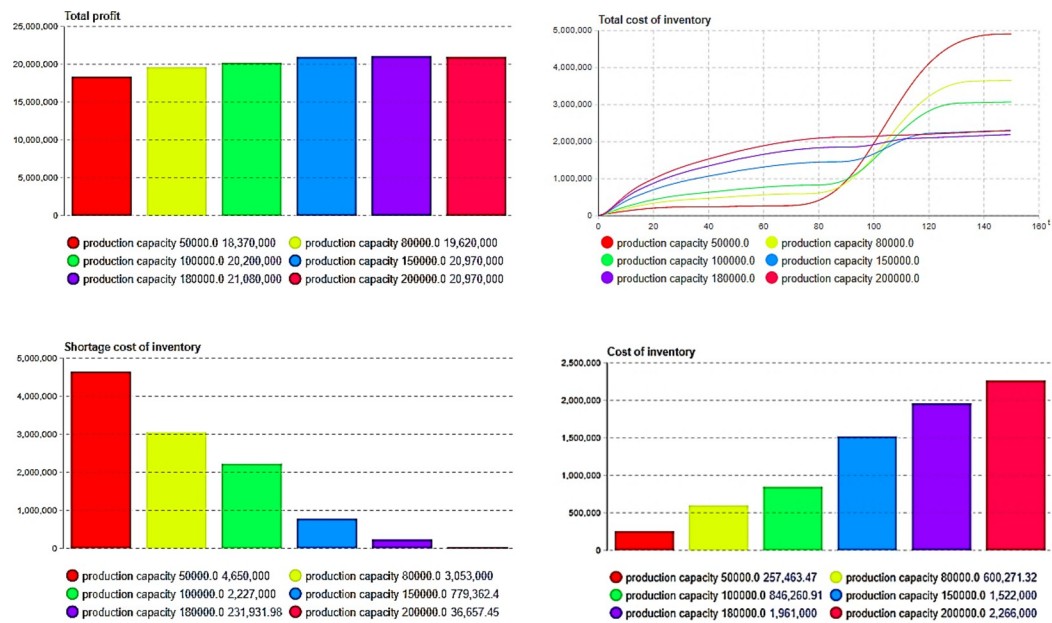

**Fig 12. The impact of production capacity on corporate profits and costs.** (a)$P_c$-total profit, (b) $P_c$-total cost, (c) $P_c$-shortage cost, (d) $P_c$-holding cost.

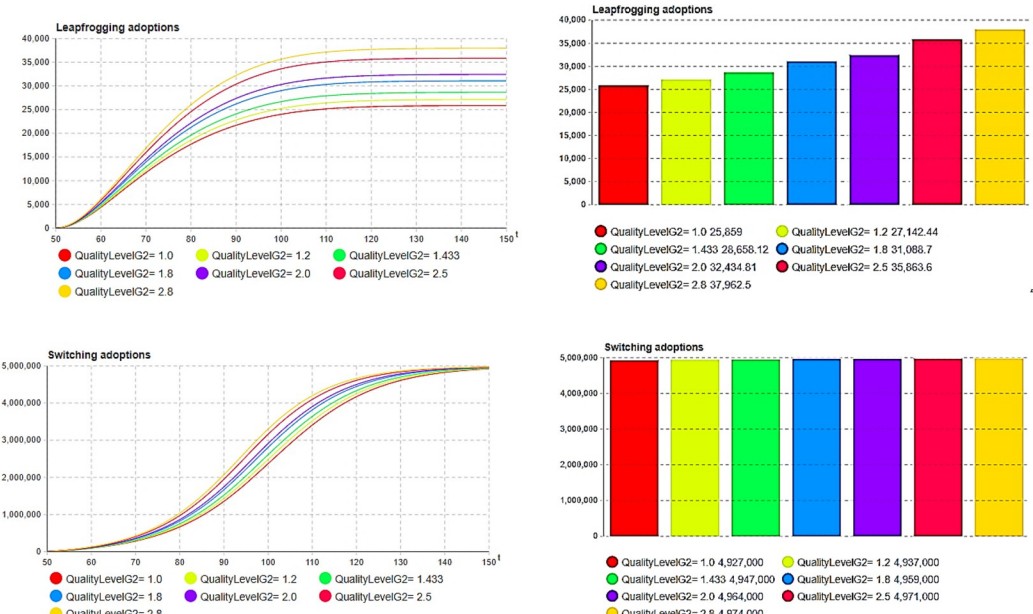

**Fig 13. The encroachment of product quality upgrade on the previous generation.** (a)Leapfrogging adoption curve ($k_1$), (b) Leapfrogging adoption rate ($k_1$), (c) Switching adoptions curve ($k_2$), (d) Switching adoptions quantity ($k_2$).

With an increase in the cumulative quantity of product sales, the market potential also decreases, which will inevitably lead to a reduction in the diffusion rate, thus resulting in a greater increase in inventory holding costs. Enterprises must optimize their production capacity to make optimal decisions.

Furthermore, the effects of the initial price and quality level on the substitution effect of multi-generation products were investigated. Fig 13(A)–13(D) illustrate a continuous increase in the number of leapfrogging buyers or adoptions and switching buyers or adoptions as a result of quality improvement upgrades. This indicates an expanding market share of second-generation products compared to their first-generation counterparts. Therefore, enterprises should carefully consider the extent of quality upgrading to avoid market erosion, which could lead to an overall profit loss. The quality level of the second-generation products has a greater impact on the leapfrog purchase, which indicates that the degree of quality upgrade of the second-generation products can have a greater impact on the sales quantity of the first-generation products. The higher the degree of quality upgrade, the greater the cannibalization effect on first-generation products. However, from the optimization results, there was an optimal value for the quality improvement level. This makes the degree of encroachment of generation products appropriate and ensures that the total profit of the enterprise reaches its maximum.

Fig 14(A)–14(D) illustrate a notable decline in the number of leapfrogging buyers or adoptions as the price of second-generation products increases. However, when $p_2(0) = 2$ (the price point is two), the number of switching buyers or adoptions peaks at approximately 4,979,000 buyers. As the price further escalates for second-generation products, encroachment on first-generation products diminishes; nevertheless, this approach fails to optimize overall company profits. It can be seen that enterprises face a trade-off between leapfrogging and switching adoptions when making price decisions for second-generation products.

Summary5: The upgrade degree of product quality and pricing level will have a greater impact on the sales share (leapfrogging adoption) of previous-generation products and a lesser impact on product-switching adoption or repeat buyers.

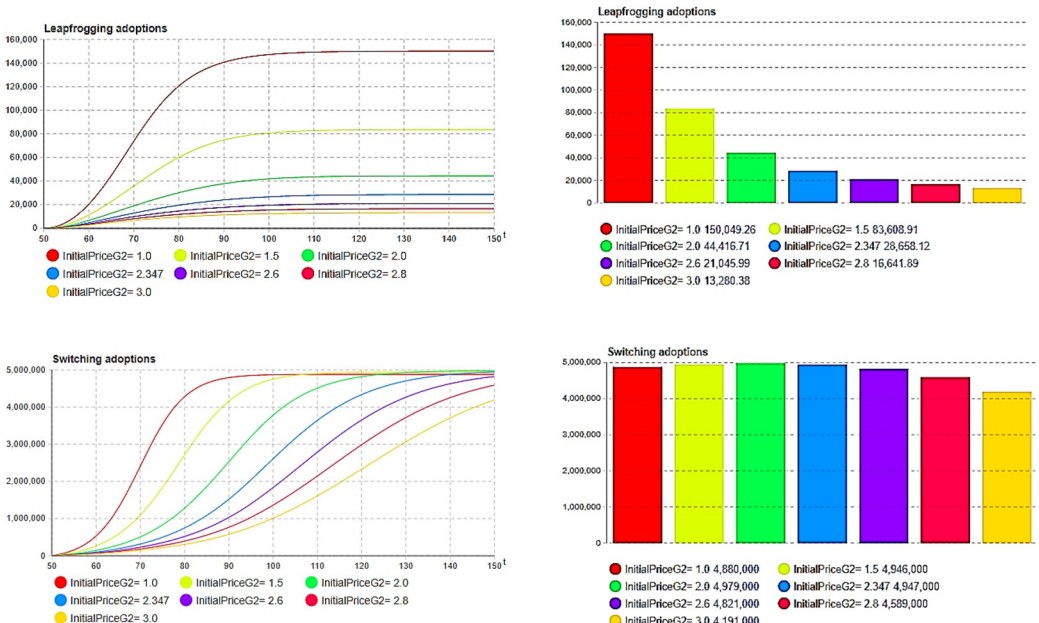

**Fig 14. The encroachment of product price levels on the previous generation.** (a) Leapfrogging adoption curve ($p_2(0)$), (b) Leapfrogging adoption rate ($p_2(0)$), (c) Switching adoptions curve ($p_2(0)$), (d) Switching adoptions quantity ($p_2(0)$).

The quality-cost coefficient can affect the production cost of a product. In the subsequent ten quality cost scenarios, optimization and calculation were performed on the system, and the results are presented in Table 5.

An increase in the quality-cost coefficient leads to a reduction in enterprise profits and a level of quality improvement for second-generation products. As shown in Table 5, the decrease in corporate profits was significantly less than the decline in quality upgrades. This is because enterprises prioritize profit maximization and choose not to improve product quality, which ultimately diminishes consumer welfare. It can also be seen that with an improvement in the quality cost level, the pricing level of the first-generation products also increases, while the pricing level of the second-generation products decreases. This indicates that the price gap between the two generations of products narrows and illustrates the correlation between pricing and the level of quality upgrade. In other words, the pricing of a new product decreases while the extent of quality improvement is comparatively lower.

We also analyzed the price sensitivity of the system. The impact of the price sensitivity coefficient on enterprises' optimal decision-making under different circumstances is presented below. System simulations were conducted, and the results are listed in Table 6.

Consumer price sensitivity directly affects the rate at which a product diffuses in the market. When the price of a product is high, consumers hesitate to purchase, resulting in delayed consumer behavior. Table 6 reveals that, as consumer price sensitivity increases, enterprises across different generations continuously lower their product pricing and experience reduced quality upgrades for second-generation products, ultimately leading to declining corporate profits. However, the reduction in consumer price sensitivity can effectively improve the total profit of the enterprise, and also the pricing level and quality upgrade level of the two generations of products. Consequently, we conclude that excessive consumer price sensitivity compels enterprises to reduce their pricing levels and weaken product-quality upgrades. Decreasing consumer price sensitivity can promote quality upgrading and improve corporate

**Table 6. Optimal decision of enterprises under different price sensitivity.**

| Parameter | $p_1(0)$ | $p_2(0)$ | $k_1$ | $k_2$ | $P_c$ | $\rho_1$ | $\rho_2$ | $\pi_{OM}$ |
|---|---|---|---|---|---|---|---|---|
| $\theta = 0.2$ | 7.387 | 9.309 | 1 | 4.121 | 205,440.295 | 0.103 | 0 | 102,866,182.872 |
| $\theta = 0.4$ | 3.702 | 4.451 | 1 | 2.471 | 202,576.09 | 0.101 | 0 | 46,984,527.73 |
| $\theta = 0.6$ | 2.465 | 2.942 | 1 | 1.824 | 196,970.4 | 0.097 | 0 | 29,607,746.115 |
| $\theta = 0.8$ | 1.84 | 2.227 | 1 | 1.471 | 184,024.718 | 0.089 | 0 | 21,223,132.855 |
| $\theta = 1$ | 1.457 | 1.78 | 1 | 1.204 | 187,487.988 | 0.091 | 0 | 16,315,257.038 |
| $\theta = 1.2$ | 1.227 | 1.537 | 1 | 1.062 | 175,434.967 | 0.084 | 0 | 13,091,325.303 |
| $\theta = 1.4$ | 1.048 | 1.332 | 1 | 1.009 | 175,046.177 | 0.083 | 0 | 10,824,856.709 |
| $\theta = 1.6$ | 0.899 | 1.147 | 1 | 1.000 | 175,574.452 | 0.083 | 0 | 9,131,493.712 |
| $\theta = 1.8$ | 0.81 | 1.082 | 1 | 1.000 | 163,946.249 | 0.077 | 0 | 7,814,747.349 |
| $\theta = 2.0$ | 0.727 | 0.972 | 1 | 1.000 | 165,277.493 | 0.077 | 0 | 6,778,068.303 |

profits; however, it is not necessarily beneficial to consumers because it makes them pay more for products.

Summary6: Price sensitivity has an important impact on enterprise decision-making. Reducing consumer sensitivity to prices can effectively promote enterprise product upgrades and improve profits.

## Discussions and practical implications

This study presents important findings that extend the theory of multi-generation product operations. The findings of this study show that the introduction of product upgrades drives companies to increase their pricing, leading them to adopt a higher-quality, higher-price strategy. At the same time, it is also necessary to consider the encroachment of second-generation products on the market share of first-generation products. In the case of a single enterprise, to pursue maximum profit, the enterprise raises its price through quality upgrades. However, certain companies engaged in multi-generation product operations may opt for relatively stable pricing due to competitive pressure [11]. An appropriate stock shortage is beneficial to the overall operation of multi-generation products in an enterprise, and this result is similar to the conclusions of the literature [34, 41]. Our findings prove that this conclusion also applies to multi-generation product diffusion. Price sensitivity has an important impact on enterprise decision-making. In the process of quality upgrading, enterprises increase their pricing level. If consumers are sensitive to price changes, the increasing pricing level will significantly affect their purchasing decision rate. In the model constructed in this study, owing to the existence of a time discount, the high price sensitivity of consumers will have a very adverse impact on corporate decision-making, and our results also show that this situation will reduce the profits of enterprises. In terms of the substitution or cannibalization effect between multi-generation products, we discuss the different influences of second-generation product pricing and quality upgrade levels on leapfrog and switching purchases. Our results show that these two marketing factors have relatively large impacts on leapfrog purchases but relatively small impacts on switching adoptions or repeat buyers. Therefore, we can conclude that the decisions of these two marketing factors have a significant impact on substitution or cannibalization between the two generations of products. Perhaps this substitution effect between products significantly affects the overall profits of the firm. Our findings further expand the literature [12] on consumer behavior in this substitution process. When production capacity is limited, the shortage becomes profoundly serious [38]. Our findings show that through comprehensive analysis and optimization of the model, the cancellation of capacity limits and the optimization of capacity

can improve corporate profits. However, shortages were inevitable. Therefore, based on the analysis of the system model, the optimal production capacity was calculated in this study.

This study has several practical implications: 1) When dealing with quality upgrades, pricing strategies for multi-generation products must carefully consider the quality levels of each generation. Enterprises should also consider the impact of second-generation products on their predecessors' market shares. 2) During the diffusion of multi-generation products, enterprises should adopt a strategy that combines price increases with quality upgrades. Consumer price sensitivity plays a pivotal role in shaping product pricing and improving product quality. Companies may need to employ strategies to mitigate consumer sensitivity to prices, thereby enhancing quality and overall profitability. 3) The cost coefficient for quality upgrades is inversely related to the degree of enhancement of multi-generation products. Enterprises should focus on improving production efficiency and product development capabilities to reduce the cost of quality upgrades, thereby improving consumer satisfaction, increasing profits, and achieving sustainable development. 4) It is crucial to recognize that a larger production capacity does not always equate to better outcomes. When determining production capacity, companies should conduct a comprehensive analysis that considers sales and inventory factors to arrive at optimal production capacity.

## Conclusions and future directions

In this study, we developed a comprehensive multi-generation product diffusion sales-supply system model with a focus on quality upgrades. This model integrated multi-generation product sales and supply, including production and inventory considerations. We adopted the system dynamics method to simulate the model in the distributed and centralized modes. We used a genetic algorithm (GA) to compute and optimize the model, enhancing the reliability of the research results by applying the Stackelberg game model of distributed decision-making in comparison with centralized decision-making.

This study unveiled the internal mechanisms governing product renewal within enterprises. We also quantified the significant impact of quality upgrades on brand influence, thereby establishing a cumulative relationship between quality upgrades and the influence of product brands. The insights derived from this study can significantly influence enterprise decision-making. Our findings indicate that in the operation of multi-generation products, maintaining a moderate level of out-of-stock situations can be beneficial for enhancing overall enterprise profits. However, companies must strike a balance between inventory costs and the expenses associated with shortages. Optimizing production capacity and allocation rates are key factors in achieving maximum profitability. Furthermore, product quality significantly influences pricing, which also affects an enterprise's total inventory costs. We also analyze and discuss the substitution effect between multi-generation products from the perspective of consumer behavior and provide the change trend of two types of consumer behavior under different pricing and quality levels of new products. It is essential to recognize that changes in various elements throughout multi-generation product operations have far-reaching implications for the overall system performance. The sales-supply system model proposed in this study expands the multi-generation product diffusion theory and also provides a decision basis and theoretical insight for the multi-generation product operation and sustainable development of enterprises.

This study primarily explored the construction and simulation of a multi-generation product sales-supply system encompassing production and inventory for a single manufacturer. Specifically, it emphasized the role of quality upgrades in this context. However, this study does not consider a competitive scenario. Future research should consider situations in competitive environments to build a novel integrated sales-production-inventory model, such as

[56]. We can also model individual behaviors and simulations in conjunction with the macro diffusion model proposed in this study. As brand, advertising, quality and other factors can affect consumers' sensitivity to product pricing. In further research, these factors can be used to construct a function of price sensitivity in the model. Given that inventories and production generate issues such as carbon emissions and carbon trading [43], this could also be factored into the model in the future. The integrated system model in this study can be further studied by constructing a differential equation control system and solving the HJB equations [53], which is a promising research direction.

## Supporting information

**S1 File.**
(ALP)

## Acknowledgments

The authors appreciate the valuable comments and suggestions of anonymous reviewers and area editors, which improved this paper both in content and representation. All authors have consented to the acknowledgement.

## Author Contributions

**Conceptualization:** Tan Bo, Kenan Yuan.

**Data curation:** Tan Bo.

**Formal analysis:** Tan Bo, Kenan Yuan.

**Funding acquisition:** Tan Bo, Kenan Yuan.

**Investigation:** Tan Bo.

**Methodology:** Tan Bo, Kenan Yuan.

**Project administration:** Kenan Yuan, Yirui Ge.

**Resources:** Kenan Yuan.

**Software:** Tan Bo.

**Supervision:** Kenan Yuan, Yirui Ge.

**Validation:** Kenan Yuan, Yirui Ge.

**Visualization:** Tan Bo.

**Writing – original draft:** Tan Bo.

**Writing – review & editing:** Tan Bo, Kenan Yuan, Yirui Ge.

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
