## [Decision Letter · Decision Letter 0]

5 Dec 2023

PONE-D-23-36882Modeling and simulating the multi-generation products sales, production and inventory system within the context of quality upgradesPLOS ONE

Dear Dr. Yuan,

Thank you for submitting your manuscript to PLOS ONE. After careful consideration, we feel that it has merit but does not fully meet PLOS ONE’s publication criteria as it currently stands. Therefore, we invite you to submit a revised version of the manuscript that addresses the points raised during the review process. Please submit your revised manuscript by Jan 19 2024 11:59PM. If you will need more time than this to complete your revisions, please reply to this message or contact the journal office at plosone@plos.org. Please include the following items when submitting your revised manuscript:A rebuttal letter that responds to each point raised by the academic editor and reviewer(s). You should upload this letter as a separate file labeled 'Response to Reviewers'.A marked-up copy of your manuscript that highlights changes made to the original version. You should upload this as a separate file labeled 'Revised Manuscript with Track Changes'.An unmarked version of your revised paper without tracked changes. You should upload this as a separate file labeled 'Manuscript'.

We look forward to receiving your revised manuscript.

Kind regards,

Hao Guo

Academic Editor

PLOS ONE

Journal Requirements:

   "National Natural Science Foundation of China, China (Grant No. 72172025); The Zhejiang Province project for planning philosophy and social sciences (Grant No. 23NDJC356YB)."

Reviewers' comments:

Reviewer's Responses to Questions

**Comments to the Author**

1. Is the manuscript technically sound, and do the data support the conclusions?

Reviewer #1: Partly

Reviewer #2: Yes

Reviewer #3: Yes

2. Has the statistical analysis been performed appropriately and rigorously? 

Reviewer #1: N/A

Reviewer #2: Yes

Reviewer #3: Yes

3. Have the authors made all data underlying the findings in their manuscript fully available?

Reviewer #1: Yes

Reviewer #2: Yes

Reviewer #3: Yes

4. Is the manuscript presented in an intelligible fashion and written in standard English?

Reviewer #1: Yes

Reviewer #2: Yes

Reviewer #3: Yes

5. Review Comments to the Author

Reviewer #1: The present paper develops a multi-generation diffusion model of a single enterprise that integrates a supply system within the framework of sustainable multi-generation product upgrading. Then this paper reveals the intricate relationship between pricing, quality level upgrade, and inventory and production capacity. I think the authors have written an interesting paper dealing with an important topic. The overall representation of this paper is technically sound. I have, however, a few comments and suggestions for them:

1. It is better to show the practical relevance of this problem setting before the problem description. It would be good if the authors could show a practical case of this problem. Moreover, it is important to show if this problem is really important.

2. The assumptions are made without the support of references. Are these assumptions practically important?

3. The literature review section is quite basic without critically reviewing existing studies and linking them with current work. The authors have just mentioned what previous authors did but they failed to describe the research gaps and relationships with the current work.

4. It is suggested to add the latest frontiers of literature and analyze their main contributions as well as their value and usefulness for this study. Please refer to-A production inventory model for high-tech products involving two production runs and a product variation. Pricing strategy in an interval-valued production inventory model for high-tech products under demand disruption and price revision.

5. A short paragraph at the end of the first section describing the manuscript structure is missing.

6. To demonstrate the true innovation of your work, consider including a chronological table of related publications at the end of the Introduction section. This will effectively highlight the hierarchy of your literature study and succinctly showcase the novelty of your work.

7. Resolution of the all the figures provided in the manuscript is too low.

8. The theoretical and practical implications of this research should be elaborated in a separate section.

9. Would it be possible to reflect and describe the limitations of the paper? And what future avenues of research do these limitations open up?

10. The authors are advised to check the English language of this manuscript. Avoid long sentences throughout the paper.

Reviewer #2: The article is based on the simulation of the model, which was conducted in a rational and scientific manner, with full disclosure of the data and procedures, and appropriate and relatively rigorous statistical analysis.

Reviewer #3: This paper analyzes the effect of new product pricing, quality levels and some other factors on the performance. Overall, the topic is interesting, the method is novel, and the results seem right. However, this paper needs more improvement before publish.

1. In Introduction, besides of the results, the authors need to show the methodology, and the contribution of this paper.

2. In Model, the exponential factors in equation 5 and 6 need more justification.

3. In Simulation and Experimentation, the way of discussion needs improvement. It is recommended to reference the way in literature: 10.1109/TSMC.2022.3191799, 10.1016/j.tre.2021.102592, 10.1371/journal.pone.0138641

Thank you. Wish to see the revised version in the near future.

6. PLOS authors have the option to publish the peer review history of their article (what does this mean?). If published, this will include your full peer review and any attached files.

Reviewer #1: No

Reviewer #2: No

Reviewer #3: No

---

## [Author Response · Author response to Decision Letter 0]

15 Jan 2024

Dear editor and reviewers:

We appreciate you and the reviewers for your precious time and hard work in reviewing our paper and providing valuable comments. It was your valuable and insightful comments that led to possible improvements in the current version. The authors have carefully considered the comments and tried our best to address every one of them. We hope the manuscript after careful revisions meet your high standards.

In response to all the proposals made in the review, we set out below all the modifications, corrections, or explanations made in the work on an item-by-item basis.

Sincerely, 

Kenan Yuan, Ph.D. 

Response to Reviewer 1 Comments

Point 1: It is better to show the practical relevance of this problem setting before the problem description. It would be good if the authors could show a practical case of this problem. Moreover, it is important to show if this problem is really important.

Response 1: First, I would like to thank the reviewers for your suggestions. We have added a case in the introduction to illustrate the research problem of this paper, and address this issue that it's really important. The contents are marked in yellow in the revised draft. The specific content is as follows begin from the line 43 to 63:

A typical example is Apple, which has advanced development and upgrading technology capabilities and also excellent marketing and supply chain operations. Apple has developed successive generations of products that are launched every year; through the joint operation of sales, production, and inventory, each generation products has always been in a hot state of purchase and obtains high profits [3]. Similar to Apple, Tesla developed a series of model products through multi-generational upgrades. Starting from Model S and progressing to Model 3, there was a gradual improvement in product performance. However, in this multi-generation product upgrade process, new product sales usually create an imbalance between strong consumer demand and Tesla's delayed delivery. Although there will be insufficient production capacity or component supply problems, this situation has repeatedly occurred. Other companies, such as Nintendo, Samsung, Xiaomi, and BYD, exhibit the same phenomenon. This recurring situation proves that these enterprises employ deliberate strategies, suggesting that new products are sold through a closely integrated approach that combines marketing strategies with supply management. Obviously, in the process of multi-generation product operations, whether the enterprise can realize the systematic analysis and planning of sales and supply of new products plays a vital strategic role in the performance improvement and sustainable growth of the enterprise. Although scholars have offered extensive research and analysis on this method of operation, most have only considered the situation of single-generation products or a subset of the multi-generation product scenario. In the case of multi-generation product upgrades, sales and supply decisions for new products will become more complicated. However, there have been insufficient studies in this area.

Point 2: The assumptions are made without the support of references. Are these assumptions practically important?

Response 2: For the questions raised by the reviewers, we have carefully considered and cited some of the contents of the hypothesis as follows:

Assumption(H2): Consumers initiate word-of-mouth discussions about a product after making a payment. As demonstrated by Shen et al. [34] in this paper, even in cases where orders face delays during the product diffusion process, consumers generate corresponding word-of-mouth feedback immediately after payment.

Assumption(H3): Because product quality serves as the foundation for product brand reputation, the quality level of the previous generation products will have an impact on brand reputation [46], which, in turn, affects the innovative adoption of the next generation products. 

Assumption(H4): Owing to constraints related to production costs and other operational expenses, we assume that prices do not decline continuously. We assume that when dynamic pricing reaches a certain point, it remains unchanged until the product is withdrawn from the market [11,26].

Assumption(H5): Referencing the literature [12], this paper assumes that consumers exhibit homogeneity in the multi-generation product diffusion process, disregarding instances of consumer heterogeneity.

The following references are cited:

34. Shen W, Duenyas I, Kapuscinski R. Optimal pricing, production, and inventory for new product diffusion under supply constraints. Manufacturing & Service Operations Management.2014; 16: 28-45.

46. Arjuna H, Ilmi S. Effect of brand image, price, and quality of product on the smartphone purchase decision. EkBis: Jurnal Ekonomi dan Bisnis. 2019; 3(2): 294-305.

11. Tan B, Zhu Z, Jiang P, Wang X. Modeling Multi-Generation Product Diffusion in the Context of Dual-Brand Competition and Sustainable Improvement. Sustainability. 2023; 15: 12920.

26. Henderson, B. The experience curve reviewed–II. History. Perspectives 1973, 125, 1–2.

12. Norton, J.A., Bass, F.M. A Diffusion Theory Model of Adoption and Substitution for Successive Generations of High-Technology Products. Management Science. 1987; 33: 1069–1086.

Point 3: The literature review section is quite basic without critically reviewing existing studies and linking them with current work. The authors have just mentioned what previous authors did but they failed to describe the research gaps and relationships with the current work.

Response 3: According to the suggestions of reviewer, we reviewed the literature again, compared it with the latest literature, and described the gap between the previous research and the current research, the content is as follows:

Although these studies have extensively researched the diffusion problem of multi-generation products, there are still significant deficiencies in modeling and simulating production and inventory within the Norton-BASS diffusion process, which expands the application scope of the Norton-Bass model in theory. Unlike Ruidas et al. [44], they mainly discuss the quality of the primary product and its impact on the demand for the pro version. We focus on a comprehensive analysis of quality upgrades in multi-generation product diffusion to make optimal sales and supply decisions. We also consider the important roles of quality and price in demand, and our research shows that quality acts on consumers through word of mouth and together with dynamic price. At the same time, insufficient attention has been paid to the characterization of the quality upgrade level and the accumulation of the quality upgrade level in these works, which have a very important effect on the brand reputation and product diffusion. To date, we found no literature that considers these aspects simultaneously.

Point 4: It is suggested to add the latest frontiers of literature and analyze their main contributions as well as their value and usefulness for this study. Please refer to-A production inventory model for high-tech products involving two production runs and a product variation. Pricing strategy in an interval-valued production inventory model for high-tech products under demand disruption and price revision.

Response 4: According to the suggestions of reviewers, we reviewed the literature again, compared it with the latest literature, and described the gap between the previous research and the current research.

Appreciate very much to the reviewers for providing us with these two excellent works, which are of great value to the improvement of the research work of this paper. We have carefully studied these two papers, and added them to the references of this paper, and have cited them.

43.Ruidas S, Seikh, M. R., Nayak, P. K. A production inventory model for high-tech products involving two production runs and a product variation. Journal of Industrial and Management Optimization. 2023; 19(3):2178-2205.

44.Ruidas S, Seikh, M. R., Nayak, P. K. Pricing strategy in an interval-valued production inventory model for high-tech products under demand disruption and price revision. Journal of Industrial and Management Optimization. 2023; 19(9):6451-6477.

The specific content is as follows:

Specifically, Ruidas et al. [43] conducted extensive and in-depth research on the multi-generation product renewal problem and proposed a model that incorporates factors such as price, advertising, demand disruption, product substitution, inventory, and production and provides the optimal inventory and pricing strategy under carbon emissions trading. Subsequently, they considered the impact of quality on an updated version of the product, briefly explained the relationship between quality and brand, and developed an EPQ model on this basis [44]. Bayrak et al. [45] considered production recycling in a closed-loop supply chain with multi-generation product diffusion; however, they did not consider the effect of inventory.

Point 5: A short paragraph at the end of the first section describing the manuscript structure is missing.

Response 5: Thanks to the suggestions of the reviewers, which make the structure of this paper more complete, we have added this part, the specific content is as follows:

The remainder of this paper is organized as follows. Section 2 reviews the literature. Section 3 describes the model and the related assumptions. Section 4 presents the model of an integrated sales and supply system. Section 5 presents the simulations and experiments of the system based on the system dynamics method. Section 6 analyzes the sensitivity of the system to variations in each parameter. In Section 7, we discuss the simulation results and the practical implications of this study. Section 8 discusses and summarizes the main contributions of this study and provides possible future research directions.

Point 6: To demonstrate the true innovation of your work, consider including a chronological table of related publications at the end of the Introduction section. This will effectively highlight the hierarchy of your literature study and succinctly showcase the novelty of your work.

Response 6: We are very grateful to the reviewers for their valuable suggestions, which have greatly improved this paper on the whole. With reference to the literature [43-44], we reviewed the literature and drew the following table to highlight the novelty and contribution of this paper.

The system model established in this study expands the application scope of the Norton-Bass model in theory and considers the management practice more diverse and comprehensively; a comparison with previous studies in the field of multi-generation product diffusion can be drawn from Table 1.

Table 1. Summary of the important Literature

Author Single/multi-generation Dynamic pricing Quality level Brand Production Inventory substitution Shortage

Shi [5] Multi √ 

Jiang [10] Multi √ √ 

Tan [11] Multi √ √ √ 

Li [24] Multi √ √ √ √ 

Kim [30] Multi √ √ 

Nadar [31] Single √ √ 

Shen [34] Single √ √ √ √ 

Kumar [35] Single √ √ √ 

Allon [36] Single √ √ √ √ 

Hartl [37] Multi √ 

Schwarz [38] Multi √ √ 

Li [39] Multi √ √ 

Negahban [40] Multi √ √ √ √ 

Bilginer [41] Multi √ √ √ 

Nagpal [42] Multi √ √ √ √ 

Ruidas [43-44] Multi √ √ √ √ √ 

Bayrak [45] Multi √ √ √ 

Our model Multi √ √ √ √ √ √ √ 

Point 7: Resolution of the all the figures provided in the manuscript is too low.

Response 7: Thanks to the reviewer for pointing out this problem in this paper. In view of the blurred figures quality raised by the reviewer, we made all the figures in the paper clear. On the basis of the original draft, the clarity of all the figures is increased by 60%.

Point 8: The theoretical and practical implications of this research should be elaborated in a separate section

Response 8: According to the reviewer's request, the theoretical and practical significance of this paper is discussed in a separate chapter as follows:

Discussions and practical implications 

This study presents important findings that extend the theory of multi-generation product operations. The findings of this study show that the introduction of product upgrades drives companies to increase their pricing, leading them to adopt a higher-quality, higher-price strategy. At the same time, it is also necessary to consider the encroachment of second-generation products on the market share of first-generation products. In the case of a single enterprise, to pursue maximum profit, the enterprise raises its price through quality upgrades. However, certain companies engaged in multi-generation product operations may opt for relatively stable pricing due to competitive pressure [11]. An appropriate stock shortage is beneficial to the overall operation of multi-generation products in an enterprise, and this result is similar to the conclusions of the literature [34,41]. Our findings prove that this conclusion also applies to multi-generation product diffusion. Price sensitivity has an important impact on enterprise decision-making. In the process of quality upgrading, enterprises increase their pricing level. If consumers are sensitive to price changes, the increasing pricing level will significantly affect their purchasing decision rate. In the model constructed in this study, owing to the existence of a time discount, the high price sensitivity of consumers will have a very adverse impact on corporate decision-making, and our results also show that this situation will reduce the profits of enterprises. In terms of the substitution or cannibalization effect between multi-generation products, we discuss the different influences of second-generation product pricing and quality upgrade levels on leapfrog and switching purchases. Our results show that these two marketing factors have relatively large impacts on leapfrog purchases but relatively small impacts on switching adoptions or repeat buyers. Therefore, we can conclude that the decisions of these two marketing factors have a significant impact on substitution or cannibalization between the two generations of products. Perhaps this substitution effect between products significantly affects the overall profits of the firm. Our findings further expand the literature [12] on consumer behavior in this substitution process. When production capacity is limited, the shortage becomes profoundly serious [38]. Our findings show that through comprehensive analysis and optimization of the model, the cancellation of capacity limits and the optimization of capacity can improve corporate profits. However, shortages were inevitable. Therefore, based on the analysis of the system model, the optimal production capacity was calculated in this study. 

This study has several practical implications: 1) When dealing with quality upgrades, pricing strategies for multi-generation products must carefully consider the quality levels of each generation. Enterprises should also consider the impact of second-generation products on their predecessors’ market shares. 2) During the diffusion of multi-generation products, enterprises should adopt a strategy that combines price increases with quality upgrades. Consumer price sensitivity plays a pivotal role in shaping product pricing and improving product quality. Companies may need to employ strategies to mitigate consumer sensitivity to prices, thereby enhancing quality and overall profitability. 3) The cost coefficient for quality upgrades is inversely related to the degree of enhancement of multi-generation products. Enterprises should focus on improving production efficiency and product development capabilities to reduce the cost of quality upgrades, thereby improving consumer satisfaction, increasing profits, and achieving sustainable development. 4) It is crucial to recognize that a larger production capacity does not always equate to better outcomes. When determining production capacity, companies should conduct a comprehensive analysis that considers sales and inventory factors to arrive at optimal production capacity.

Point 9: Would it be possible to reflect and describe the limitations of the paper? And what future avenues of research do these limitations open up?

Response 9: Thanks for the reviewer's suggestion. In the last paragraph of the paper, the limitations and future research directions of this study are show as follows:

This study primarily explored the construction and simulation of a multi-generation product sales-supply system encompassing production and inventory for a single manufacturer. Specifically, it emphasized the role of quality upgrades in this context. However, this study does not consider a competitive scenario. Future research should consider situations in competitive environments to build a novel integrated sales-production-inventory model, such as [56]. We can also model individual behaviors and simulations in conjunction with the macro diffusion model proposed in this study. As brand, advertising, quality and other factors can affect consumers' sensitivity to product pricing. In further research, these factors can be used to construct a function of price sensitivity in the model. Given that inventories and production generate issues such as carbon emissions and carbon trading [43], this could also be factored into the model in the future. The integrated system model in this study can be further studied by constructing a differential equation control system and solving the HJB equations [53], which is a promising research direction.

Point 10: The authors are advised to check the English language of this manuscript. Avoid long sentences throughout the paper.

Response 10: We have reorganized the whole content of the article to avoid using long sentences. In addition, according to the editor's suggestion, we have found professionals to revise the grammar and sentences of the article.

Response to Reviewer 2 Comments

Point 1: In order to strengthen your literature review and theoretical implications, you may want to include more recent and relevant references published in recent months (years), which currently appear to be farther back in time for this paper.

Response 1: Thanks to the reviewer for this valuable suggestion. According to the reviewer's suggestions. We have added several recent references, these are as follows:

42. Nagpal G., Chanda U. Optimal inventory policies for short life cycle successive generations’ technology products. Journal of Management Analytics. 2022; 9(2): 261-286

43.Ruidas S, Seikh, M. R., Nayak, P. K. Pricing strategy in an interval-valued production inventory model for high-tech products under demand disruption and price revision. Journal of Industrial and Management Optimization. 2023; 19(9):6451-6477.

44.Ruidas S, Seikh, M. R., Nayak, P. K. A production inventory model for high-tech products involving two production runs and a product variation. Journal of Industrial and Management Optimization. 2023; 19(3):2178-2205.

54.Wang X, Sethi, S.P., Chang S. Pollution abatement using cap-and-trade in a dynamic supply chain and its coordination. Transportation Research Part E: Logistics and Transportation Review. 2022; 158: 102592.

56. Wang X, Zhang S. The interplay between subsidy and regulation under competition. IEEE Transactions on Systems, Man, and Cybernetics: Systems. 2022; 53(2): 1038-1050.

Point 2: Try to illustrate your findings in the literature review section by comparing it with the existing literature. The literature review section should be presented logically and try to explain the more novel ideas in your study compared to the existing literature. Provide a table of author contributions in this section to highlight the unique novelty of this paper.

Response 2: Thank the reviewers for their valuable suggestions, we have carefully sorted out the literature, and the contents revised and added are as follows begin from the line 209:

Although these studies have extensively researched the diffusion problem of multi-generation products, there are still significant deficiencies in modeling and simulating production and inventory within the Norton-BASS diffusion process, which expands the application scope of the Norton-Bass model in theory. Unlike Ruidas et al. [44], they mainly discuss the quality of the primary product and its impact on the demand for the pro version. We focus on a comprehensive analysis of quality upgrades in multi-generation product diffusion to make optimal sales and supply decisions. We also consider the important roles of quality and price in demand, and our research shows that quality acts on consumers through word of mouth and together with dynamic price. At the same time, insufficient attention has been paid to the characterization of the quality upgrade level and the accumulation of the quality upgrade level in these works, which have a very important effect on the brand reputation and product diffusion. To date, we found no literature that considers these aspects simultaneously.

The system model established in this study expands the application scope of the Norton-Bass model in theory and considers the management practice more diverse and comprehensively; a comparison with previous studies in the field of multi-generation product diffusion can be drawn from Table 1.

Table 1. Summary of the important Literature

Author Single/multi-generation Dynamic pricing Quality level Brand Production Inventory substitution Shortage

Shi [5] Multi √ 

Jiang [10] Multi √ √ 

Tan [11] Multi √ √ √ 

Li [24] Multi √ √ √ √ 

Kim [30] Multi √ √ 

Nadar [31] Single √ √ 

Shen [34] Single √ √ √ √ 

Kumar [35] Single √ √ √ 

Allon [36] Single √ √ √ √ 

Hartl [37] Multi √ 

Schwarz [38] Multi √ √ 

Li [39] Multi √ √ 

Negahban [40] Multi √ √ √ √ 

Bilginer [41] Multi √ √ √ 

Nagpal [42] Multi √ √ √ √ 

Ruidas [43-44] Multi √ √ √ √ √ 

Bayrak [45] Multi √ √ √ 

Our model Multi √ √ √ √ √ √ √ 

Point 3: The profit function in the model section is confusing, try to explain which are costs, which are revenues and which are profits. Equations (7-9) confuse me as well, if the first derivative in equation (7) should be dS_1(t)/dt as stated in the text?

Response 3: For the questions raised by the reviewers, we make the following explanations. In these equations, π_srepresents revenue from sales, H_c represents production and inventory costs, and π_OM represents overall operating profit.

The two subsystems were merged to create a sale-supply system as follows, and the total profit of the new system is expressed as π_OM=π_s-H_c. This content is added at line 343 in the paper.

Thanks to the reviewer for pointing out these errors for us, it was indeed an oversight on our part and has been corrected. In equation (7), correct writing is indeed dS1(t)/dt

Point 4: The authors should add a "real" discussion section with extensive comments on the results obtained. I would like to know how the results obtained in this paper are similar to or different from other studies, which are currently unclear.

Response 4: According to the reviewer's suggestions, we add a discussion section, the theoretical and practical significance of this paper is discussed in a separate chapter as follows:

Discussions and practical implications 

This study presents important findings that extend the theory of multi-generation product operations. The findings of this study show that the introduction of product upgrades drives companies to increase their pricing, leading them to adopt a higher-quality, higher-price strategy. At the same time, it is also necessary to consider the encroachment of second-generation products on the market share of first-generation products. In the case of a single enterprise, to pursue maximum profit, the enterprise raises its price through quality upgrades. However, certain companies engaged in multi-generation product operations may opt for relatively stable pricing due to competitive pressure [11]. An appropriate stock shortage is beneficial to the overall operation of multi-generation products in an enterprise, and this result is similar to the conclusions of the literature [34,41]. Our findings prove that this conclusion also applies to multi-generation product diffusion. Price sensitivity has an important impact on enterprise decision-making. In the process of quality upgrading, enterprises increase their pricing level. If consumers are sensitive to price changes, the increasing pricing level will significantly affect their purchasing decision rate. In the model constructed in this study, owing to the existence of a time discount, the high price sensitivity of consumers will have a very adverse impact on corporate decision-making, and our results also show that this situation will reduce the profits of enterprises. In terms of the substitution or cannibalization effect between multi-generation products, we discuss the different influences of second-generation product pricing and quality upgrade levels on leapfrog and switching purchases. Our results show that these two marketing factors have relatively large impacts on leapfrog purchases but relatively small impacts on switching adoptions or repeat buyers. Therefore, we can conclude that the decisions of these two marketing factors have a significant impact on substitution or cannibalization between the two generations of products. Perhaps this substitution effect between products significantly affects the overall profits of the firm. Our findings further expand the literature [12] on consumer behavior in this substitution process. When production capacity is limited, the shortage becomes profoundly serious [38]. Our findings show that through comprehensive analysis and optimization of the model, the cancellation of capacity limits and the optimization of capacity can improve corporate profits. However, shortages were inevitable. Therefore, based on the analysis of the system model, the optimal production capacity was calculated in this study. 

Point 5: Please use the correct styles for all formula inputs and tables in the article.

Response 5: We are very grateful to the reviewers for their valuable suggestions. We have corrected styles for all formula inputs and tables in the article and fixed some errors in notation. And we have checked and adjusted all the formulas in the paper.

Response to Reviewer 3 Comments

Point 1: In Introduction, besides of the results, the authors need to show the methodology, and the contribution of this paper. 

Response 1: Thanks to the suggestions of the reviewers, we have elaborated the methods and contributions of this study in more detail in the paper as follows at line 101 to 109 in the paper:

The main purpose of this research is to develop a multi-generation diffusion model of a single enterprise that integrates a supply system within the framework of sustainable multi-generation product upgrading, then explore the pricing, production capacity and inventory decisions of multi-generation products under the context of quality levels upgrading, and maximize overall profits as the goal of the simulations. This paper reveals the intricate relationship between pricing, quality level upgrade, inventory and production capacity; expands the multi-generation products diffusion theory; and provides a theoretical basis for enterprises’ new product launches and operation decisions. In one word, this research explores the path to continuous product innovation and improvement for enterprises.

Point 2: In Model, the exponential factors in equation 5 and 6 need more justification. 

Response 2: According to the reviewer's suggestions, we add arguments to this part, the contents are as follows:

Based on equation (3) and equation (4), we add the price and quality factors. Reference [10,34], we add pricing as a marketing factor to the diffusion process of products, its expression is e^(-〖θp〗_1 (t) ). The diffusion of first-generation products is affected by their quality level k_1though word of mouth, while the diffusion of second-generation products is not only affected by their own quality level k_2, but also by the quality level of the previous generation, that is, the degree of quality upgrade, that is, k_1-k_2. It affects second-generation products through its cumulative effect on the brand [46], and the degree of improvement in quality can be expressed by (k_2-k_1)/k_1 .

Point 3: In Simulation and Experimentation, the way of discussion needs improvement. It is recommended to reference the way in literature: 10.1109/TSMC.2022.3191799, 10.1016/j.tre.2021.102592, 10.1371/journal.pone.0138641

Response 3: First of all, we are very grateful to the reviewer for the three excellent literatures, which offer a great help for our work. Although we are not sure whether we have the ability to meet the standards of these excellent works, we have modified them as much as possible to meet the reviewer's requirements. On the basis of referring to and citing these literatures, we have carefully revised our paper and added some explanations and contents to the simulation and experiment parts. The specific contents have been marked in yellow in the revised draft. In addition, these papers also bring inspiration to our future work. In the following work, we are preparing to use the optimal control theory to study the competition of multi-generation products for enterprises. These literatures have pointed out the direction for our research work, especially the fitted finite volume solution of HJB equations, which is very needed for our future work and add these contents to the future directions in our paper. Thanks again for reviewer.

53. Chang S, Wang X, Wang Z. Modeling and computation of transboundary industrial pollution with emission permits trading by stochastic differential game. PLoS ONE. 2015; 10: e0138641.

54. Wang X, Sethi, S.P., Chang S. Pollution abatement using cap-and-trade in a dynamic supply chain and its coordination. Transportation Research Part E: Logistics and Transportation Review. 2022; 158: 102592.

56. Wang X, Zhang S. The interplay between subsidy and regulation under competition. IEEE Transactions on Systems, Man, and Cybernetics: Systems. 2022; 53(2): 1038-1050.

For example, in the paper, we add the following content:

As shown in Fig 8(b), when the initial price of the first-generation product reaches the optimum, that is, p_1 (0)=1.825, the profit of the current-generation products also reaches a maximum, and the change trend also shows an increase and then a decrease.

These contents are at line552 to line 555.

It is worth noting that the initial inventory of the second-generation products can be reduced to a very low level, or even “zero stock,” which indicates that through dynamic production and replenishment, the overall profit of enterprises can be improved. 

These contents are at line 572 to line 574.

The quality level of the second-generation products has a greater impact on the leapfrog purchase, which indicates that the degree of quality upgrade of the second-generation products can have a greater impact on the sales quantity of the first-generation products. The higher the degree of quality upgrade, the greater the cannibalization effect on first-generation products. However, from the optimization results, there was an optimal value for the quality improvement level. This makes the degree of encroachment of generation products appropriate and ensures that the total profit of the enterprise reaches its maximum.

These contents are at line 604 to line 610.

It can also be seen that with an improvement in the quality cost level, the pricing level of the first-generation products also increases, while the pricing level of the second-generation products decreases. This indicates that the price gap between the two generations of products narrows and illustrates the correlation between pricing and the level of quality upgrade. In other words, the pricing of a new product decreases while the extent of quality improvement is comparatively lower.

These contents are at line 634to line 639.

Given that inventories and production generate issues such as carbon emissions and carbon trading [43], this could also be factored into the model in the future. The integrated system model in this study can be further studied by constructing a differential equation control system and solving the HJB equations [53], which is a promising research direction.

These contents are at line 736 to line739.

Finally, I would like to appreciate the reviewers for the valuable comments on this paper, which have further improved quality of this paper. Thanks again to the reviewers.

---

## [Editor Report · Decision Letter 1]

20 Feb 2024

Modeling and simulating the multi-generation product sales, production and inventory system within the context of quality upgrades

PONE-D-23-36882R1

Dear Dr. Yuan,

We’re pleased to inform you that your manuscript has been judged scientifically suitable for publication and will be formally accepted for publication once it meets all outstanding technical requirements.

Kind regards,

Hao Guo

Academic Editor

PLOS ONE
---

## [Editor Report · Acceptance letter]

19 Mar 2024

PONE-D-23-36882R1 

PLOS ONE

Dear Dr. Yuan, 

I'm pleased to inform you that your manuscript has been deemed suitable for publication in PLOS ONE. Congratulations! Your manuscript is now being handed over to our production team.

Kind regards, 

on behalf of

Dr. Hao Guo 

Academic Editor

PLOS ONE